# Neonatal genetics of gene expression reveal potential origins of autoimmune and allergic disease risk

Qin Qin Huang [1,2,3✉], Howard H. F. Tang[1,4], Shu Mei Teo[1,5], Danny Mok[6], Scott C. Ritchie [1,5,7,8], Artika P. Nath[1,5], Marta Brozynska[1,5], Agus Salim[9,10,11], Andrew Bakshi [12], Barbara J. Holt[6], Chiea Chuen Khor [13,14,15], Peter D. Sly [16], Patrick G. Holt [6,16], Kathryn E. Holt [17,18] & Michael Inouye [1,2,5,7,8,19,20,21✉]

Chronic immune-mediated diseases of adulthood often originate in early childhood. To investigate genetic associations between neonatal immunity and disease, we map expression quantitative trait loci (eQTLs) in resting myeloid cells and CD4[+] T cells from cord blood samples, as well as in response to lipopolysaccharide (LPS) or phytohemagglutinin (PHA) stimulation, respectively. *Cis*-eQTLs are largely specific to cell type or stimulation, and 31% and 52% of genes with *cis*-eQTLs have response eQTLs (reQTLs) in myeloid cells and T cells, respectively. We identified *cis* regulatory factors acting as mediators of *trans* effects. There is extensive colocalisation between condition-specific neonatal *cis*-eQTLs and variants associated with immune-mediated diseases, in particular *CTSH* had widespread colocalisation across diseases. Mendelian randomisation shows causal neonatal gene expression effects on disease risk for *BTN3A2*, *HLA-C* and others. Our study elucidates the genetics of gene expression in neonatal immune cells, and aetiological origins of autoimmune and allergic diseases.

[1] Cambridge Baker Systems Genomics Initiative, Baker Heart and Diabetes Institute, Melbourne, VIC 3004, Australia. [2] Department of Clinical Pathology, University of Melbourne, Parkville, VIC 3010, Australia. [3] Department of Human Genetics, Wellcome Sanger Institute, Cambridge, UK. [4] School of BioSciences, The University of Melbourne, Parkville, VIC 3010, Australia. [5] Cambridge Baker Systems Genomics Initiative, Department of Public Health and Primary Care, University of Cambridge, Cambridge CB1 8RN, UK. [6] Telethon Kids Institute, The University of Western Australia, Perth, WA 6009, Australia. [7] British Heart Foundation Cardiovascular Epidemiology Unit, Department of Public Health and Primary Care, University of Cambridge, Cambridge, UK. [8] National Institute for Health Research Cambridge Biomedical Research Centre, University of Cambridge and Cambridge University Hospitals, Cambridge, UK. [9] Baker Heart and Diabetes Institute, Melbourne, VIC 3004, Australia. [10] School of Mathematics and Statistics, The University of Melbourne, Parkville, VIC 3010, Australia. [11] Melbourne School of Population and Global Health, Carlton, VIC 3053, Australia. [12] Monash Biomedicine Discovery Institute, Prostate Cancer Research Group, Department of Anatomy and Developmental Biology, Monash University, Clayton, VIC 3800, Australia. [13] Human Genetics, Genome Institute of Singapore, Agency for Science, Technology and Research, Singapore 138672, Singapore. [14] Singapore Eye Research Institute, Singapore, Singapore. [15] Duke-NUS Medical School, Singapore, Singapore. [16] Child Health Research Centre, The University of Queensland, Brisbane, QLD 4101, Australia. [17] Department of Infectious Diseases, Central Clinical School, Monash University, Melbourne, VIC 3004, Australia. [18] The London School of Hygiene and Tropical Medicine, London WC1E 7TH, UK. [19] The Alan Turing Institute, London, UK. [20] British Heart Foundation Centre of Research Excellence, University of Cambridge, Cambridge, UK. [21] Health Data Research UK Cambridge, Wellcome Genome Campus and University of Cambridge, Cambridge, UK. ✉email: qh1@sanger.ac.uk; minouye@baker.edu.au

Infancy is a critical period during which physiological and developmental changes impact the pathogenesis of conditions later in life[1,2]. Many complex diseases, in particular immune and respiratory conditions, are partially determined by genetic predisposition and early-life environment exposures, such as microbes or allergens[3–5]. Yet, despite increasing evidence of its importance, little is known about the early-life genetic regulation of gene expression, nor its relevance to predisposition for diseases in adulthood.

Expression quantitative trait loci (eQTL) studies have provided insights into the gene regulatory effects of genetic variants and their relationship with complex disease[6,7]. The majority of eQTLs have been identified in adult tissues, while eQTLs in perinatal tissues have only been explored recently: for example, eQTLs identified in foetal placentas[8] and foetal brains[9] are enriched for genetic variants associated with growth (e.g. adult height) and neuropsychiatric (e.g. schizophrenia) traits, respectively. In addition to genetic variation, disease development is influenced by individual and cell-type-specific responses to external stimuli. Understanding the interaction between eQTLs and these stimuli can give insights into condition(s), whether they be cell type, microbe or temperature, under which genetic variants may influence disease. Previous studies have investigated response eQTLs (reQTLs), eQTLs with genetic effects modified by external stimulation, in CD14$^+$ monocytes[10,11], macrophages[12], dendritic cells[13,14], and CD4$^+$ T cells[15]. However, reQTL studies to date have been largely performed using adult samples, and we currently have limited knowledge of how neonatal immune gene expression is regulated in response to stimuli.

Here, we characterise the genetics of gene expression in the innate and adaptive arms of the neonatal immune system using purified cord blood samples from 152 neonates[16–18]. In these samples, we identify cis- and trans-eQTLs of myeloid cells and CD4$^+$ T cells, as well as reQTLs for myeloid cells stimulated with lipopolysaccharide (LPS; a component of bacterial cell walls) and CD4$^+$ T cells stimulated with phytohemagglutinin (PHA; a pan-T cell mitogen). By comparing our neonatal cis-eQTLs with previously identified adult eQTLs, we find eQTLs that are specific to neonates. Using mediation analysis, instances of putative trans gene regulation are investigated to identify cis regulatory mechanisms. We show evidence for the shared genetic basis of neonatal eQTLs and reQTLs with common autoimmune and allergic diseases, and many of such colocalisations are cell type- or stimulation-specific. Finally, we use Mendelian randomisation to uncover the causal effects of neonatal gene expression on risk of immune-mediated diseases. Ultimately, we highlight the potential importance of the perinatal period in understanding the origins of immune-mediated disease.

## Results

### Genetics of neonatal gene expression in innate and adaptive immunity.
We performed eQTL analysis on cell preparations derived from in vitro cultures of resting and stimulated neonatal immune cells from 152 neonates of the Childhood Asthma Study (CAS) cohort (Fig. 1), which were enriched respectively for non-adherent T cells and adherent myeloid cells as detailed in 'Methods', with the latter largely being monocytes and macrophages. Cell purities could not be experimentally confirmed by flow cytometry due to limited blood volumes that could be collected from neonates; rather, in silico analyses were utilised to estimate the abundances of relevant cell types using CIBERSORTx[19]. These analyses indicated dominance of the relevant T-cell and myeloid signatures (Supplementary Fig. 1). Among 135 genotyped individuals, 106 and 119 had gene expression data passing QC for both resting and stimulated conditions of myeloid

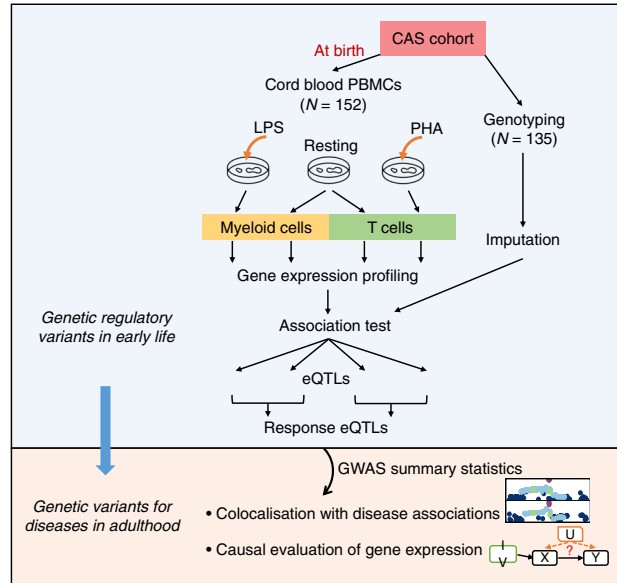

**Fig. 1 Study design and analysis work flow.** Monocyte/macrophage-enriched cultures (myeloid cells) and T-cell-enriched cultures were extracted from resting and stimulated cord blood samples from the Childhood Asthma Study (CAS) cohort. Gene expression was quantified using a microarray platform. Genotype data are available for a subset of the CAS individuals. eQTLs were identified within each experimental condition. Datasets for resting and stimulated samples were merged to detect response eQTLs within each cell type. Next, we identified genetic loci where neonatal eQTLs and disease associations obtained from external GWAS datasets shared the same causal variants. We investigated the causal effects of gene expression at birth on immune diseases that develop later in life.

and T-cell cultures, respectively, and 95 individuals had post-QC data for all four cultures. The total number of samples available for eQTL analysis was 116 for resting myeloid cells, 125 for LPS-stimulated myeloid cells, 126 for resting T cells, and 127 for PHA-stimulated T cells.

To identify cis-eQTLs, we applied a hierarchical procedure to correct for multiple testing within each experimental condition at 5% false discovery rate (FDR; 'Methods'). Stimulated cells yielded a larger number of cis-eQTLs and associated genes (eGenes) than resting cells (1347 vs. 971 eGenes in PHA-stimulated vs. resting T cells, respectively; 376 vs. 136 in LPS-stimulated vs. resting myeloid cells, respectively; Fig. 2a, Supplementary Data 1). To investigate the differences in numbers of eGenes between conditions, we repeated the analysis controlling for differences in sample size (randomly sampling 116 samples in each condition). This yielded similar results to the numerical distribution of cis-eGenes: 1231, 900, and 350 in PHA-stimulated T cells, resting T cells, and LPS-stimulated myeloid cells, respectively. The lower number of eQTLs in myeloid cells may be explained by fewer genes being expressed (Supplementary Fig. 2).

For eGenes with eQTLs in multiple experimental conditions, we performed conditional analysis to distinguish whether these were independent or shared signals between conditions ('Methods'). The majority (74%) of eQTL signals were specific to one cell type or stimulatory condition (Supplementary Fig. 3A), consistent with previous observations[10]. About 10–50% of the condition-specific signals were replicated using a multivariate adaptive shrinkage (mash) model (Supplementary Fig. 3B)[20]. We observed a majority of cis-eQTL effects after stimulation: 60% (262 of 376) of eGenes in LPS-stimulated myeloid cells and 58%

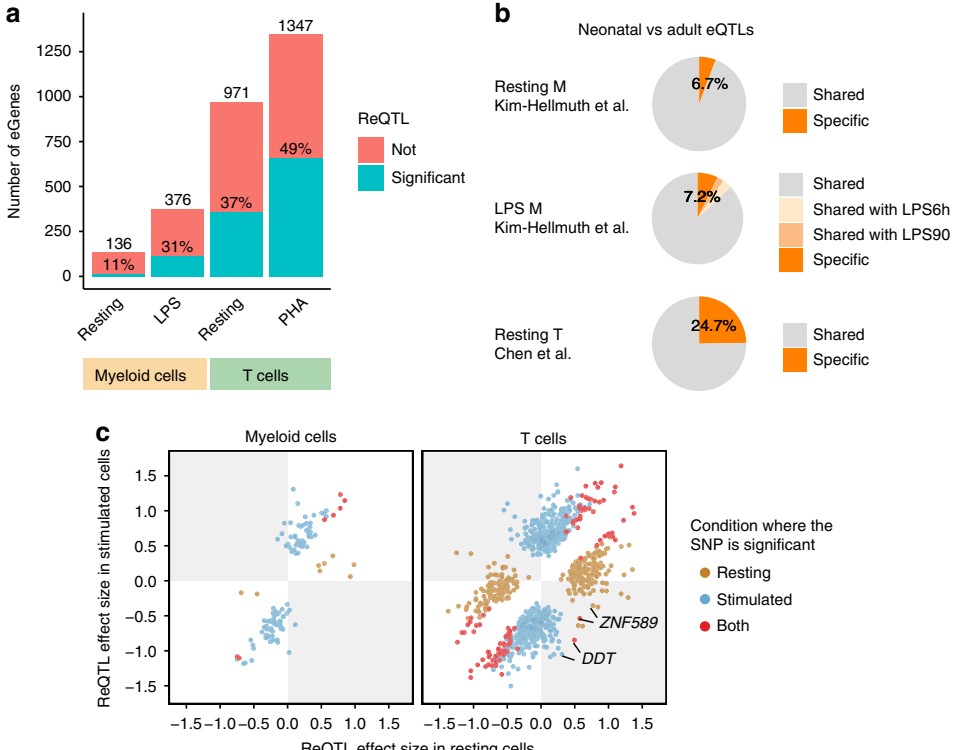

**Fig. 2 *Cis*-eQTLs and response eQTLs (reQTLs) in myeloid cells and T cells. a** A bar plot shows the number of genes with significant *cis*-eQTLs (eGenes) at a 5% false discovery rate (FDR) identified in each cell type and treatment group (on *x*-axis). Percentages on each bar indicate the proportion of eGenes with significant reQTLs at 5% FDR ('Methods'). **b** Three pie charts show the proportions of eQTLs that were specific to neonates in resting myeloid cells (*n* = 116 biologically independent samples), LPS-stimulated myeloid cells (*n* = 125), and resting T cells (*n* = 126), respectively. Neonatal specific eQTLs were defined by comparing the eQTLs in our study with that identified in resting and LPS-stimulated (90 min or 6 h) monocytes obtained from adults (Kim-Hellmuth et al.[11], *n* = 134), and that in naïve adult T cells (BLUEPRINT project[22], *n* = 125; 'Methods'). Myeloid eQTLs that were not tested in the Kim-Hellmuth et al. study (31 and 44 in resting and stimulated conditions, respectively) or T-cell eQTLs that were not tested in the Chen et al. study (332 in resting T cells) were excluded in the analysis. **c** Two point plots show effect sizes (difference in gene expression in s.d. per allele) of significant reQTLs in resting (*x*-axes) and stimulated conditions (*y*-axes) in two cell populations: myeloid cells (left) and T cells (right). A gene might have two dots indicating two independent top SNPs ('Methods'). Colours indicate the condition in which the SNP was significant. ReQTLs of *DDT* and *ZNF585* in the grey quadrants (red dots) show opposite directions of eQTL effects across conditions.

(778 of 1347) in PHA-stimulated T cells. Using a two-step conditional analysis ('Methods'), PHA-stimulated T cells had the largest number of eGenes (6.3%; Supplementary Table 1) with multiple independent eQTL signals. *GARFIELD* enrichment analysis[21] showed that the *cis*-eSNPs were enriched in 3′ untranslated regions (UTR), 5′ UTR, and exon regions (Supplementary Fig. 4), consistent with known mechanisms of *cis*-eQTLs.

We compared our resting and LPS-stimulated myeloid cells with those from adults in Kim-Hellmuth et al.[11] using the mash model[20]. We found that 6.7% and 7.2% of the *cis*-eQTLs tested in both studies (105 and 332 eQTLs) were specific to neonates at LFSR of 0.05 (Fig. 2b). Similarly, we compared resting neonatal T cells to adult T cells of the BLUEPRINT project[22]: after excluding 332 *cis*-eQTLs that were not tested in the adult T cells, we found that 24.7% neonatal *cis*-eQTLs were not active in adult T cells. We have also used the Storey and Tibshirani *q*-value approach[23] to investigate eQTL sharing between neonatal and adult tissues. Replication rate quantified by $\pi_1$ statistic was 0.998 for resting myeloid cells, 0.987 and 0.947 for LPS-stimulated myeloid cells comparing to monocytes treated by LPS for 90 min and 6 h, respectively, in Kim-Hellmuth et al., and 0.879 for resting T cells. We note, however, that this approach may overestimate the replication rate as it does not take the direction or magnitude of eQTL effects into account.

**Genetics of neonatal gene expression in response to stimuli.** To quantify how genetic regulation of gene expression is altered by external stimuli, we identified response eQTLs (reQTLs) and response eGenes (reGenes) by performing interaction tests on the top eSNPs of each eGene in myeloid cells and T cells separately, and controlling FDR at 5% using permutation-determined P-values ('Methods'). In myeloid cells, we identified 125 significant reQTLs involving 125 unique reGenes (31% of 398 myeloid eGenes); in T cells, we identified 956 reQTLs involving 918 unique reGenes (52% of 1749 T cell eGenes), among which 38 reGenes had distinct *cis*-eQTLs in two conditions where both eQTLs were reQTLs (Supplementary Data 2). Consistent with our findings for *cis* eQTLs and eGenes, the number of reQTLs and proportion of reGenes were greater in stimulated compared to resting conditions.

For two reQTLs, the direction of eQTL effect changed between conditions (Fig. 2c). The C allele of the top eSNP (rs5751775) for *DDT* (D-dopachrome tautomerase), a gene functionally related to the inflammatory cytokine *MIF* (migration inhibitory factor), increased *DDT* transcription in resting T cells but decreased expression after PHA stimulation (Supplementary Fig. 5A, Supplementary Data 2). Similarly, the T allele of the top eSNP (rs13068288) for *ZNF589* increased transcription of *ZNF589* in resting T cells but reduced expression after PHA stimulation (Supplementary Fig. 5B, Supplementary Data 2).

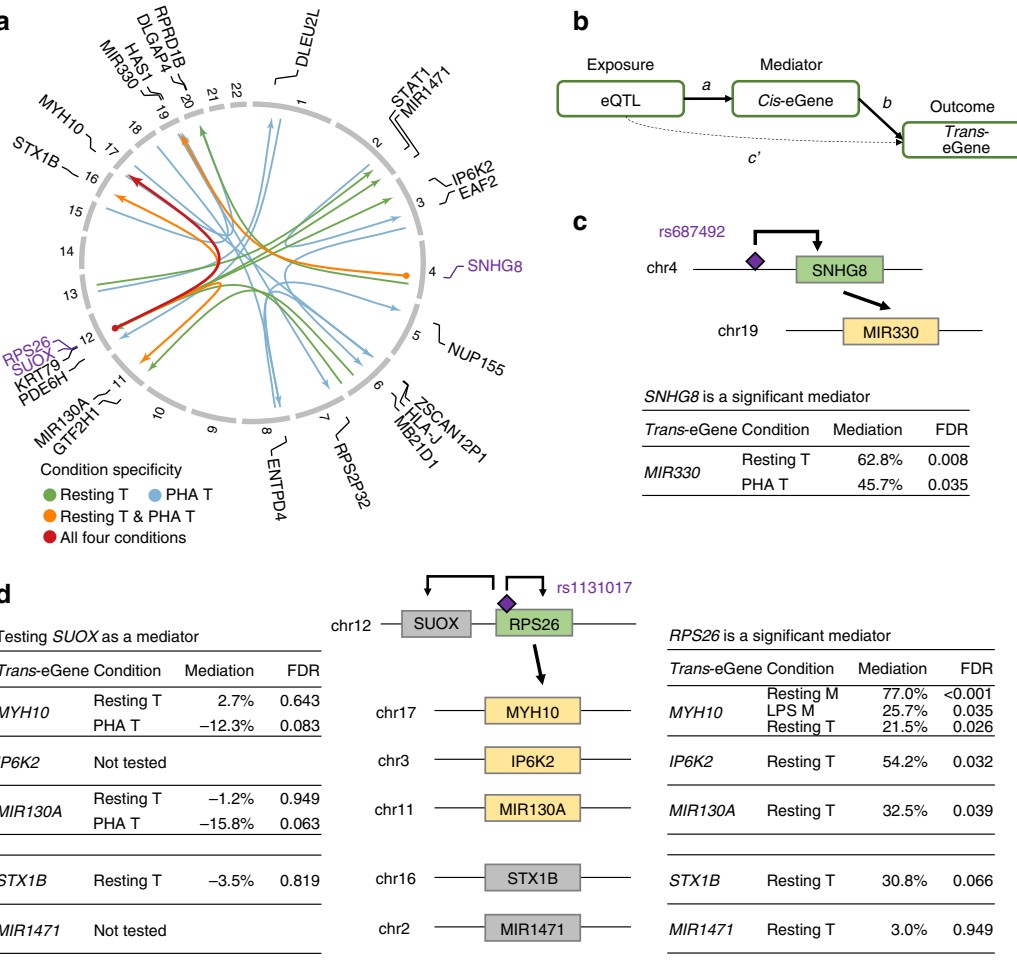

**Fig. 3 Trans-eQTL effects and their cis-mediators. a** A circular plot shows trans-eQTL associations in lines, with arrows pointing to trans-eGenes with names annotated (in black) outside the rim indicating chromosome numbers. Dots on the other end point indicate nearby genes (names in purple) that are associated with the same loci (cis-eQTLs). Colours of the lines indicate the experimental conditions where the trans-eQTLs were identified: resting T cells only (Resting T), stimulated T cells only (PHA T), shared between both conditions of T cells (Resting T & PHA T), and shared across all four experimental conditions (All four conditions). **b** A diagram demonstrates the mediation analysis model, where effects of trans-acting eQTL (exposure) on trans-eGene (outcome) are either mediated through a cis-eGene (mediator), or through direct effects ('Methods'). **c, d** Show two examples of cis-eGenes (green), SNHG8 and RPS26, acting as mediators for trans-effects (trans-eGenes in yellow). Genes that were not significant in mediation analysis are in grey. Tables show statistics of the mediation tests, and the column named Mediation indicates the proportion of total effects of the eQTL on the trans-eGene that was mediated through the cis-eGene. Two models involving SUOX (**d**) were not tested because the trans-eSNPs of IP6K2 and MIR1471 were not significantly associated with SUOX (Supplementary Data 3). Significant mediations (FDR ≤ 0.05) are highlighted in bold.

**Disentangling trans and cis effects using mediation analysis.** While we were relatively underpowered to detect trans-eQTLs, we maintained stringent significance thresholds and compared overlapping cis and trans effects to generate hypotheses of gene regulation ('Methods'). We identified 25 trans-eQTLs in T cells (10 in resting, 15 in PHA-stimulated), and one trans-eQTL in myeloid cells (Fig. 3a, Supplementary Data 3) at a genome-wide FDR of 5%. Notably, the trans-eQTL for MYH10, a component of myosin heavy chain which regulates cytokinesis, was shared across all four experimental conditions; furthermore, the same eQTL was associated with multiple trans-eGenes in T cells: MIR130A and STX1B in resting and stimulated T cells, and IP6K2 and MIR1471 in resting T cells only (Fig. 3a).

Consistent with previous reports that trans-eQTLs are enriched for cis-eQTLs[24], we found that multiple trans-eQTLs (the single trans-eQTL in myeloid cells, 6 of the 10 in resting T cells, and 3 of the 15 in PHA-stimulated T cells) were significantly associated with local genes in cis (Fig. 3a, Supplementary Data 3). The trans-eQTL for MYH10 was also a cis-eQTL for RPS26 (part of the 40S subunit of the ribosome) in all conditions except PHA-stimulated

T cells. The top eSNP (rs1131017) for RPS26, located in its 5' UTR, was also a cis-eQTL for SUOX (sulphite oxidase, a homodimer in the intermembrane space of mitochondria) in resting and stimulated T cells. Separately, the trans-eQTL (rs687492) for microRNA MIR330 was also a cis-eQTL for the long non-coding RNA SNHG8 in resting and stimulated T cells.

Mediation analysis revealed that the trans-eQTL effects of rs687492 on MIR330 were cis mediated through SNHG8 (Fig. 3b, c, Supplementary Table 2; 'Methods'), indicating a potential pathway containing this lncRNA-miRNA cross-talk. Furthermore, mediation analysis also revealed the regulatory logic of cis-eQTL (rs1131017) for SUOX and RPS26, identifying that its trans-effects on MYH10, MIR130A, and IP6K2 were mediated through RPS26 and not SUOX in resting T cells (Fig. 3d, Supplementary Table 2).

**Genetic overlap with immune-mediated diseases.** To investigate the genetic overlap between neonatal gene expression and disease, we used a multi-pronged approach. First, we performed GARFIELD enrichment analyses to test for significant overlaps

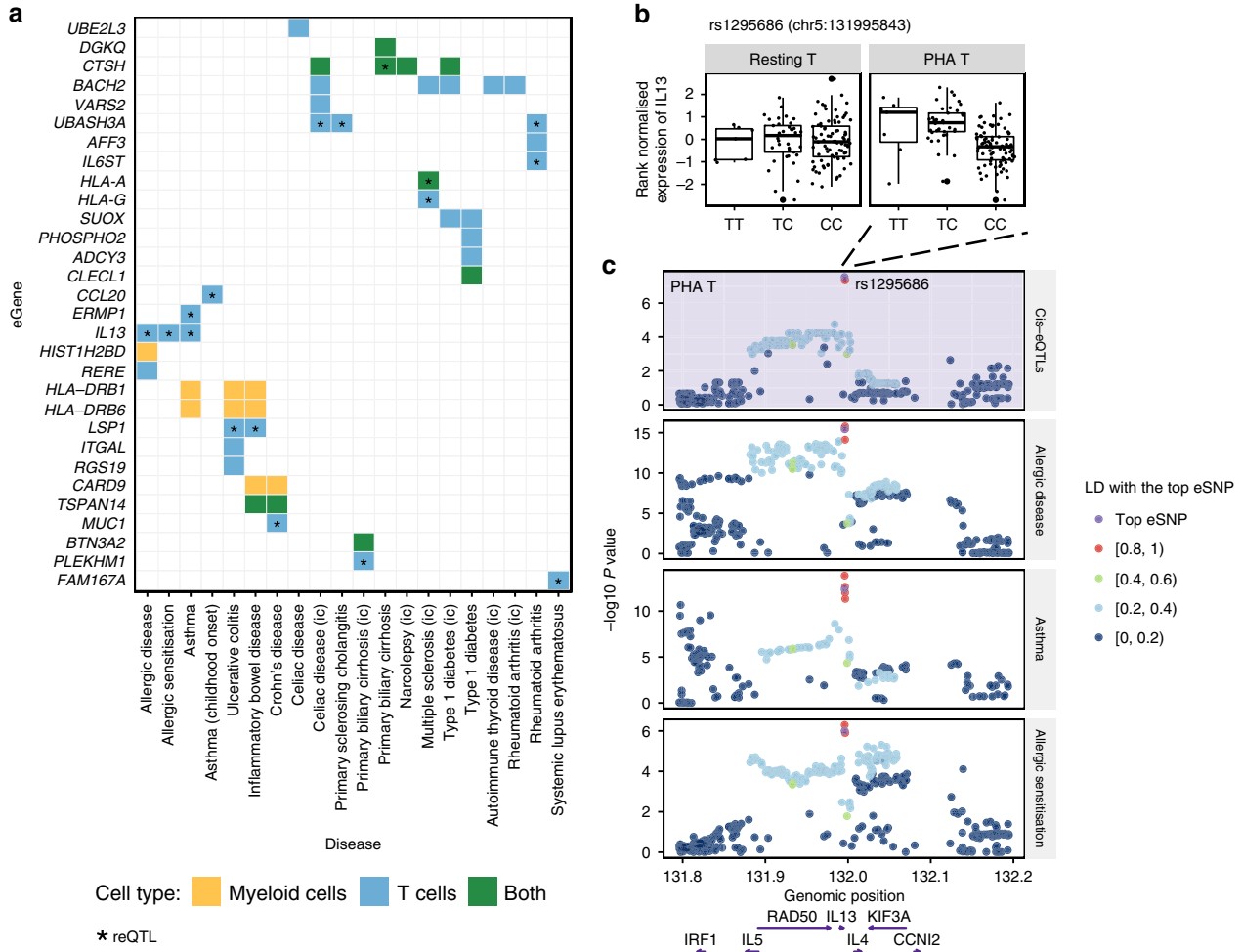

**Fig. 4 Colocalisation of *cis*-eQTLs with disease associations. a** A heatmap shows all cases with strong evidence of colocalisation between *cis*-eQTLs of corresponding genes (eGenes) in rows and GWAS hits associated with allergic and autoimmune diseases in columns (ic: the study was performed using ImmunoChip array). Colours indicate the cell type where the significant colocalisation was observed. Asterisks indicate that the colocalised eQTLs are response eQTLs (reQTLs). **b** Boxplots show the rank-normalised gene expression of *IL13* (*y*-axes) in resting T cells (left) and in PHA-stimulated T cells (right) stratified by genotypes of the reQTL rs1295686 (*x*-axes), the top eSNP in PHA-stimulated T cells. The centre line corresponds to median, and lower and upper hinges indicate the 25th and 75th percentiles. The upper whisker extends from the hinge to the largest value ≤ 1.5 * inter-quartile range (IQR) from the hinge, and the lower whisker extends from the hinge to the smallest value at most 1.5 * IQR of the hinge. The highest and lowest dots show the maximum and minimum gene expression. In resting T cells, no SNP was significantly associated with *IL13*. **c** Regional plots show eQTL association with gene expression of *IL13* in PHA-stimulated T cells (purple background), and GWAS associations with allergic disease (asthma, hay fever, or eczema), asthma, and allergic sensitisation. The minus log10 *P*-value is plotted on *y*-axes for all SNPs located within 200 kb from the top eSNP of *IL13*. Colours of dots indicate the LD correlation with the top eSNP (in purple). Positions of genes located on this locus are shown at the bottom.

between the *cis*-eQTLs and variants associated with immune-mediated diseases in genome-wide association studies (GWAS; 'Methods'). We found widespread enrichment amongst *cis*-eQTLs for genetic variants associated with diseases such as allergic disease (asthma, hay fever, or eczema) and inflammatory bowel disease. Conversely, same analysis with non-immune-related traits such as educational attainment identified limited enrichment (Supplementary Fig. 6). Specifically, we did not observe significant enrichment of GWAS signals for educational attainment in myeloid eQTLs; there was evidence of enrichment in resting and PHA-stimulated T cells but enrichment was not as strong (effect sizes 0.43 and 0.52 respectively) compared to immune-mediated diseases (ranging from 1.13 to 7.37).

Second, we performed colocalisation analysis[25] to identify variants sharing regulatory (eQTL) and disease-associated (GWAS) signals ('Methods'). In total, we observed 68 colocalisations, involving 5, 9, 15, and 17 independent *cis*-eQTLs in resting myeloid cells, LPS-stimulated myeloid cells, resting T cells, and

PHA-stimulated T cells, respectively (Fig. 4a, Supplementary Data 4). Our analysis replicated the colocalisation of the *cis*-eQTL for *BACH2* in resting T cells with variants for type 1 diabetes[22], and also revealed widespread colocalisation with autoimmune thyroid disease, celiac disease, multiple sclerosis, rheumatoid arthritis (Supplementary Fig. 7). *BACH2* encodes a transcriptional repressor that restrains terminal differentiation and promotes the development of memory lymphocytes including CD8[+] T cells[26] and B cells[27]. At the *BACH2* locus, the A allele at the top eSNP (rs72928038) was associated with decreased *BACH2* expression and increased risk of the above diseases. This was consistent with previous studies which showed that mutations and loss-of-function variants of *BACH2* resulted in immunodeficiency and disruption to regulatory T cell function, with subsequent autoimmunity[28,29].

We found 17 colocalisations of reQTLs and disease variants, in total involving 12 reQTLs: one myeloid reQTL specific to LPS stimulation (eQTL for *CTSH*), and 11 reQTLs in T cells, among

which eight were specific to PHA stimulation. Notably, the reQTL for *IL13* in PHA-stimulated T cells colocalised with GWAS hits associated with asthma, allergic sensitisation, and allergic disease (Fig. 4, Supplementary Data 4). The T allele of the top eSNP (rs1295686) was associated with greater *IL13* expression in PHA-stimulated T cells as well as increased risk of all three diseases (Fig. 4b). rs1295686 is intronic to *IL13* and in strong LD ($r^2 > 0.98$) with four other eSNPs, including a Gln144Arg missense SNP (rs20541) in *IL13*. At the *CCL20* locus, the A allele of the *cis*-eQTL/reQTL (rs13034664) in PHA-stimulated T cells was associated with lower *CCL20* expression as well as increased risk of childhood-onset asthma (Fig. 4a, Supplementary Fig. 8, Supplementary Data 4). *CCL20* is part of the *CCR6-CCL20* receptor-ligand axis, a key driver of dendritic cell chemotaxis[30].

Our analyses uncovered complex condition-specific colocalisations at multiple loci. The ubiquitin ligand *UBASH3A*, known to regulate apoptosis in T cells, had two independent *cis*-eQTLs in resting and stimulated T cells which were also reQTLs. However, only the *cis*-eQTL in resting T cells (rs1893592) colocalised with celiac disease, rheumatoid arthritis, and primary sclerosing cholangitis (PSC; Fig. 4, Supplementary Data 4). *Cis*-eQTLs for *CTSH*, which encodes the lysosomal cysteine proteinase cathepsin H, colocalised with signals for different diseases in a cell type- and condition-specific manner (Supplementary Fig. 9, Supplementary Data 4). *Cis*-eQTLs for *CTSH* in resting myeloid cells and resting T cells both colocalised with GWAS hits for celiac disease, narcolepsy, and type 1 diabetes, which has been observed in immune cell types from adults: colocalisation with causal variants of celiac disease and narcolepsy was reported in macrophages[12], and type 1 diabetes in adult monocytes[22]. On the other hand, *cis*-eQTLs for *CTSH* in LPS-stimulated myeloid cells and PHA-stimulated T cells both colocalised with primary biliary cirrhosis (PBC).

**Causal effects of condition-specific gene expression on immune-mediated diseases**. To identify putative causal effects of neonatal gene expression on risk of autoimmune and allergic disease, we performed two-sample Mendelian randomisation (MR) analysis using *cis*-eQTLs as genetic instruments, the neonatal *cis*-eGene as exposure, and disease as outcome ('Methods'). We tested the 52 eGenes which had three or more genetic instruments available, and the diseases above, for which we had GWAS summary statistics available. We considered genes for which at least three of four MR methods (inverse variance weighted, weighted median, weighted mode, and MR Egger) were in agreement in detecting significant causal effects ($P$-value $\leq 0.05$) on a disease without significant pleiotropic effects (Supplementary Data 5).

In our MR analysis, we found multiple conditions where neonatal gene expression had a causal effect on multiple diseases (Fig. 5), including *BTN3A2* (butyrophilin subfamily 3 member A2), *HLA-C* (major histocompatibility complex class I molecule), *MICB* (ligand for an activatory receptor expressed on natural killer cells, CD8$^+$ αβ T cells, and γδ T cells), *ZNRD1* (RNA polymerase 1 subunit), and *SLC22A5* (carnitine transporter) (Supplementary Data 5). *BTN3A2* had a relatively large number of genetic instruments for resting (seven to eight) and stimulated (three to five) T cells, and the causal estimates were similar between these two conditions (Supplementary Figs. 10, 11). In resting T cells, increased expression of *BTN3A2* was causally associated with decreased risk of asthma (weighted mode causal estimate = −0.056 log odds decrease per s.d. increase in *BTN3A2*), both childhood- and adult-onset asthma (−0.047 and −0.039, respectively), allergic rhinitis (−0.044), PSC (−0.440), and systemic lupus erythematosus (SLE; −0.256). Conversely,

increased *BTN3A2* expression was associated with increased risk of inflammatory bowel disease (IBD; 0.025), including Crohn's disease (0.053), as well as risk of PBC (0.129), where PBC variants also showed colocalisation with *BTN3A2* eQTLs (Fig. 4, Supplementary Fig. 12). Expression of *HLA-C* in T cells showed strong causal association with autoimmunity, in particular positive causal effects on psoriasis, SLE, PSC, multiple sclerosis, IBD, and ulcerative colitis; and negative causal effects on juvenile idiopathic arthritis, PBC, and rheumatoid arthritis (Fig. 5, Supplementary Figs. 13, 14).

## Discussion

In this study, we investigated the genetic regulation of gene expression in cells of neonatal innate and adaptive immunity, and its relationship to the genetic basis of autoimmune and allergic diseases. In this context, we illustrated that the genetics of gene expression in neonates is strongly specific to cell type and stimulatory condition, and is distinct from that of adults. We described regulatory mechanisms of eQTLs whose putative *trans* effects had evidence of mediation via gene expression in *cis*. In exploring the potential early-life origins of disease, our analyses showed an extensive genetic overlap of genetic variants associated with immune-mediated diseases and those with effects on gene expression in neonatal immune cells. Finally, Mendelian randomisation showed that myriad changes in gene expression at birth had potentially causal effects on autoimmune and allergic disease risk.

We observed stimuli changing the direction of eQTL effects in resting and PHA-stimulated T cells. This was the case for the C allele of reQTL (rs5751775) and *DDT*, a cytokine structurally and functionally related to *MIF*, a critical regulator of both innate and the adaptive immune response[31]. It is known that eQTL effects can change direction in immune cells[32]. In the Genotype-Tissue Expression (GTEx) Project[6], the direction of eQTL effect for the C allele at rs5751775 was also variable across liver, pancreas, stomach, testis, brain, and muscle tissues, though the variable direction of eQTL effect may also be due to LD contamination, i.e. when the top eSNPs tag two distinct eQTLs in two conditions.

Our results suggest a *trans* mediation role for *RPS26*, rather than *SUOX*, consistent with previous studies linking *RPS26* to *IP6K2*[33,34]. *RPS26* and *SUOX* were both identified as significant mediators in multiple tissues in the GTEx dataset; however, within the same tissue, the *trans*-associations mediated through *SUOX* were not identical to those through *RPS26*, indicating distinct effects of *SUOX* and *RPS26* on distant genes[34]. *RPS26* encodes a ribosomal subunit protein which, apart from its role in ribosome assembly and translation[35], is involved in various other cellular processes, including nonsense-mediated mRNA decay[36] and p53 transcriptional activity[37]. It is likely that the broad *trans* effects of *RPS26* may be related to its ribosomal functions.

At the *CCL20* locus, the reQTL (rs13034664) in PHA-stimulated T cells colocalised with childhood-onset asthma (Fig. 4a, Supplementary Fig. 8). *CCL20* encodes a C–C chemokine ligand that binds to a G protein-coupled receptor, and elevated CCL20 expression has been shown in airways of patients with chronic obstructive pulmonary disease (COPD)[38] and asthma[39]. CCL20 induces mucin production by binding to its unique receptor (CCR6) in human airway epithelial cells[40]. However, in PHA-stimulated T cells, the A allele of the reQTL (rs13034664), which was linked to increased risk of childhood-onset asthma, was associated with reduced *CCL20* expression (Supplementary Fig. 8B). This reQTL and its direction of effect were replicated by others in activated CD4$^+$ T cells[7]. T cells themselves respond to CCL20 via binding to CCR6, and this process is observed during

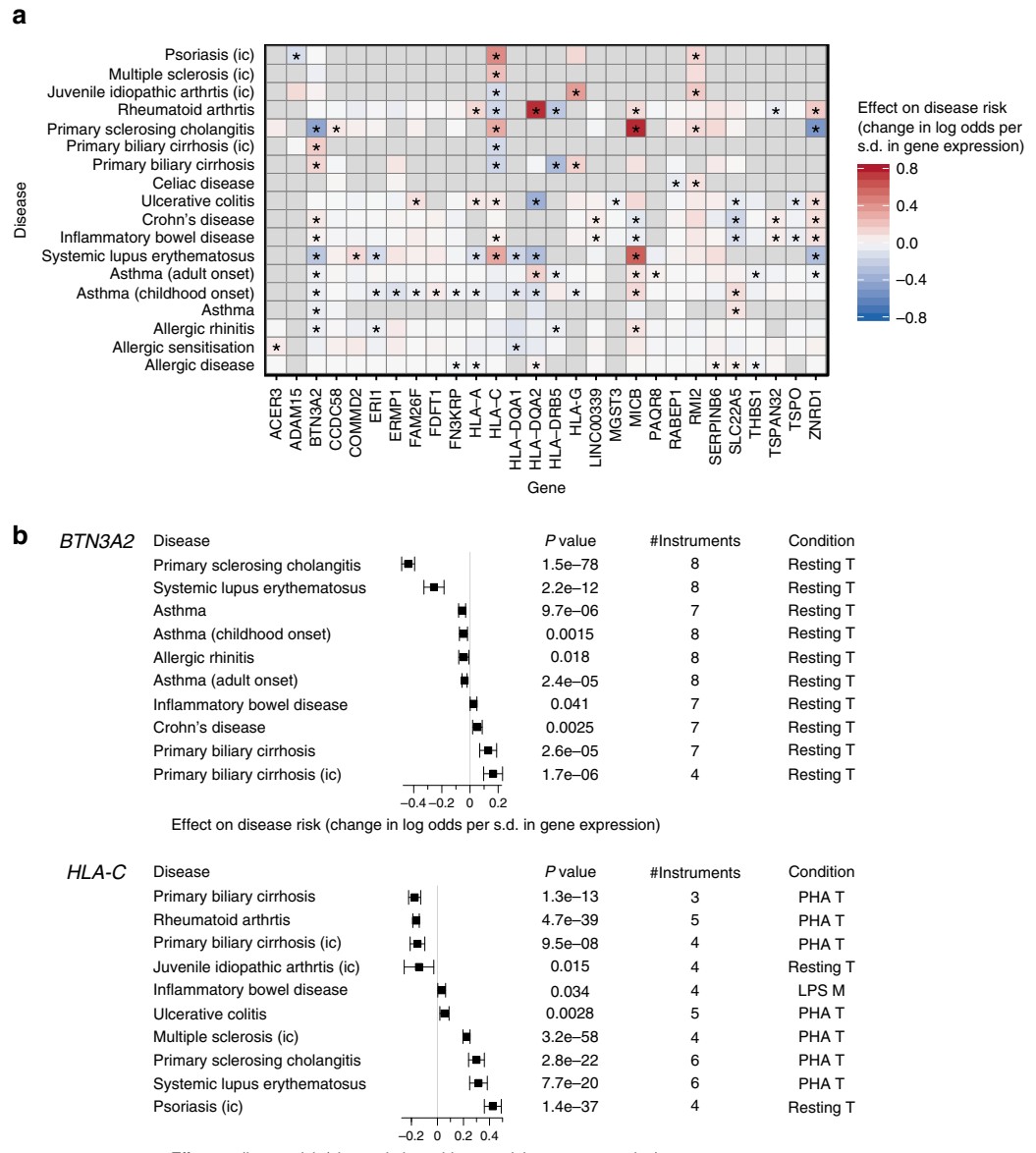

**Fig. 5 Causal effects of neonatal gene expression on multiple immune-related diseases.** The causal effects were estimated using the weighted mode method in the Mendelian randomisation (MR) analysis in both panels. Significant causal associations were defined as having $P$-value $\leq 0.05$ in at least three out of four MR methods ('Methods'). Multiple testing correction was not applied. Causal associations with significant pleiotropy were excluded. If a gene was tested using the expression levels in multiple experimental conditions, the one with the highest number of genetic instruments was kept. Statistics of all MR tests are in Supplementary Data 5. **a** Asterisks indicate significant causal associations. Grey indicates the gene-disease pairs that were not tested due to small number of genetic instruments (<3). Positive effect estimates in red indicate that increased gene expression is causally associated with increased disease risk, and negative causal associations are in blue. **b** Forest plots present estimated causal effects on disease risk and its 95% confidence intervals for neonatal expression of *BTN3A2* and *HLA-C*, using 125, 126, and 127 biologically independent samples from LPS-stimulated myeloid cells, resting T cells, and PHA-stimulated T cells.

allergen provocation[41]. It is possible that the pathophysiological mechanism is not T cell production of CCL20, but rather their response to it – hence the reduced *CCL20* expression in a possible feedback loop. Altogether, these findings reinforce the cell-type-specific nature of eQTL and reQTL associations.

On the other hand, the reQTL for *IL13* in neonatal PHA-stimulated T cells appeared to share causal variants with allergic disease and asthma, suggesting that this reQTL may affect allergic disease risk via mechanisms that do involve T cell activation and interleukin 13 (IL-13). PHA is a pan-T cell mitogen, and downstream intracellular signalling may be shared between PHA and allergen-mediated T cell activation. IL-13 is produced by activated

CD4[+] and CD8[+] T cells[42] among others, promoting immunoglobulin E (IgE) production in B cells[43]. IL-13 has been shown to induce asthma symptoms including airway hyper-responsiveness, increased total serum IgE, and increased mucus production in murine models[44]. Increased IL-13 expression is observed in sputum and bronchial biopsy in mild[45] and severe[46] asthma and can serve as a biomarker for severe refractory asthma[47]. Therapies that target IL-13 (anti-IL-13 antibodies) have been developed, such as lebrikizumab and tralokinumab however, they show inconsistent or only modest effects in treating severe asthma exacerbations in phase 3 clinical trials. Our results are consistent with IL-13 as a therapeutic target for asthma and suggest the

potential for improved efficacy if treatment is targeted towards specific genetic subgroups.

We found strong causal effects of neonatal *BTN3A2* expression on various autoimmune and allergic diseases. Butyrophilin (BTN) family members are immunoglobulin-like molecules that act as immune check-point regulators with roles in self-tolerance[48]. Increased BTN3A2 protein expression is a favourable prognostic biomarker in epithelial ovarian cancer patients, and indicates a higher density of intraepithelial infiltration of T cells[49]. BTN3 family members *BTN3A1* and *BTN3A3* are proximal to *BTN3A2*. The antigen-presenting BTN3A1 is critical to human γδ T cell activation[50]. A recent study showed that BTN3A2 regulated subcellular localisation of BTN3A1, and both were required for T cell activation[51]. Previous MR analysis found *BTN3A2* lung expression had a causal effect on COPD risk[52]. Our findings suggest that altered neonatal *BTN3A2* expression, with presumed subsequent dysfunction in immunomodulation, plays a role in the pathogenesis of multiple inflammatory conditions.

In our study, neonatal *HLA-C* expression in both myeloid cells and T cells was causally associated with multiple autoimmune diseases such as psoriasis, SLE, and primary biliary cirrhosis. *HLA-C* encodes an MHC Class I receptor which presents antigens to CD8+ T cells. It is also the major ligand for killer immunoglobulin-like receptors (KIRs), which regulate the activity of natural killer (NK) cells. *HLA-C* is an established locus for psoriasis susceptibility[53] and the interaction between *HLA-C* and *ERAP1* is associated with psoriasis risk, where *ERAP1* variants only have psoriasis effects in individuals with the *HLA-C* risk allele[54]. In our analysis of resting neonatal T cells, the largest causal effect of *HLA-C* expression was for psoriasis.

Our study had limitations. Cord blood may be under the influence of in utero exposures (e.g. smoking, drug exposures, maternal stress) which may confound associations with maternal, placental or foetal genetics[55], and may affect neonatal gene expression through epigenetic mechanisms[56]. In utero exposures, epigenetics, or maternal genetics were not measured in this cohort. While maternal cells are present in cord blood samples, the vast majority of cord blood cells are from neonates; previous estimates of the proportion of maternal cells in cord blood are of the order of $10^{-4}$ to $10^{-5}$ of nucleated foetal blood cells[57]. While the adherence protocol used in this study can lead to some level of cell activation, this is also the case with other cell isolation methods such as MACS, and there is no a priori reason to suspect that this would deleteriously influence subsequent responsiveness. Since this is a study in neonates, the blood volumes collected were necessarily small and mandated the use of micro methods for in vivo studies where all available material was needed to generate transcriptional profiles, and ancillary assays, e.g. quantification of cell purity, were not able to be performed. While RNA-seq technology allows the identification of additional signals, such as isoform QTLs, which are relevant in neonatal tissues, it can be prone to artefacts when constructing large-scale libraries with low input RNA; therefore, as this neonatal study necessarily relied on low sample volumes, we consciously chose gene expression arrays due to their robustness in this setting and well-established normalisation approaches. In silico analyses to infer cell type abundances showed good cell purities (Supplementary Fig. 1), and the estimated abundances were correlated with the PEER factors included when mapping eQTLs; thus, while differences in cell composition may not be completely controlled, we believe these effects are minimal. Furthermore, we are still relatively power-limited with regard to *trans*-eQTL analyses and further studies with greater sample sizes, particularly in the neonatal and early childhood settings, are necessary.

In conclusion, our study shows the remarkable complexity of the genetic regulation of gene expression in the innate and adaptive arms of the immune system at birth, and its potential role in the pathogenesis of autoimmunity and allergic disease.

## Methods

**Study cohort and RNA sample preparation.** The study population is a subset of the CAS, a prospective birth cohort of 234 individuals followed from birth to up to 10 years of age[16–18]. This project was approved by the University of Western Australia Human Ethics Committee. Cord blood samples were collected for 152 individuals at birth, and we have obtained informed consent from all participants. One million peripheral blood mononuclear cells (PBMCs) from each individual were stimulated with either an innate immune system stimulant (LPS: lipopoly-saccharide; 25 pg/ml), or a pan T cell stimulant (PHA: phytohemagglutinin; 1 µg/ml) for 24 h (Fig. 1). Initial stimulation prior to cell enrichment was carried out in unfractionated PBMCs to enable interactions to occur between myeloid/lymphoid cells to mimic in vivo conditions in which stimulation events occur in mixed cell environment. Unstimulated resting PBMC samples were also available. At the end of the culture period, non-adherent cells in suspension from resting and PHA-stimulated cultures were aspirated gently and enriched for CD4+ T cells (enriched T cells) using Dynabeads (Invitrogen) and stored in RNAprotect Cell reagent (Qiagen). Cells remaining in suspension in resting and LPS-stimulated cultures were aspirated, leaving an enriched population of monocytes and macrophages (enriched myeloid cells) adhered to the culture wells, and these were also resuspended vigorously into RNAprotect. Cell purities were not experimentally confirmed by flow cytometry because of the limited cell numbers available, and instead as a compromise to estimate the abundances of relevant cell types from the resultant gene expression data using CIBERSORTx[19]. All cells in RNAprotect Cell reagent were banked at −80 °C.

The cells were thawed and centrifuged briefly for RNA extraction. Reagent was removed and total RNA was extracted from pelleted cells by an established in-house procedure using TRIzol (Life Technologies) in combination with RNEasy MinElute columns (Qiagen). The aqueous phase containing the RNA was then loaded onto an RNeasy MinElute column (Qiagen) to purify and concentrate the RNA. RNA quality was assessed on a Bioanalyzer 2100 using the RNA 6000 Nano kit (Agilent). There were 607 samples (one missing sample) in total from 152 individuals for gene expression profiling.

PCA of gene expression data showed the four clusters of samples of different cell types and conditions (Supplementary Fig. 15). For both deconvolution analysis (using web-based CIBERSORTx) and PCA, we used gene expression data that were quantile normalised across all 557 samples that passed QC (see the next section for the QC and normalisation pipeline), and the plots show the 494 samples from individuals with genotype data available that were included in the eQTL analysis.

**Gene expression profiling and data processing.** Total RNA from four cell culture conditions (resting and LPS-stimulated myeloid cells, and resting and PHA-stimulated T cells) was quantified with Illumina HumanHT-12 v4 BeadChip gene expression array at the Genome Institute of Singapore. After excluding 31 samples with suspected cross-contamination or insufficient quantity of cDNA, 576 samples were successfully scanned. Supplementary Fig. 16 shows our QC and normalisation pipeline details. The raw microarray data and probe detection *P*-values were exported by the Illumina software GenomeStudio. We first removed three samples with zero intensity for almost all probes including negative controls and probes targeting housekeeping genes (two resting myeloid samples and one LPS-stimulated myeloid sample). We further removed 16 outlier samples (eight resting myeloid samples, five LPS-stimulated myeloid samples, and three resting T cell samples) with a low number of detectable probes (lying outside median ±2 × inter quartile range). Compared with other samples, these excluded samples had much lower intensity for positive controls including those targeting housekeeping genes.

After quality control, 557 samples remained for normalisation (resting monocyte/macrophage-enriched cultures: 130, LPS-stimulated monocyte/macrophage-enriched cultures: 141, resting T-cell-enriched cultures: 142, and PHA-stimulated T-cell-enriched cultures: 144). We performed background correction based on the intensity of the 770 negative control probes on the microarray, and then we performed quantile normalisation and log2 transformation within each cell type and condition using the *neqc* function from the *limma* (v3.36.5) R package[58]. We used updated probe annotation data and restricted the analysis to 33,436 reliable probes, excluding unaligned probes and probes aligned to multiple regions that were more than 25 bp apart[59]. Fifteen probes with missing data in ≥5 samples were removed. Detectable probes targeting autosomal genes ($N = 20,532$) were kept, comprising of probes with GenomeStudio detection *P*-values ≤ 0.01 in ≥2.5% of the samples from a specific condition group, or in ≥5% of all samples[10]. Gene annotation was obtained from the GENCODE release 19 (GRCh37 alignment, downloaded in October 2017). Among detectable probes, 19,230 had gene annotation in the GENCODE reference data. For genes that had multiple probes, we kept the probe with the highest mean intensity, resulting in 13,109 autosomal genes. For eQTL analysis, we performed a rank-based inverse normal transformation within each group, so that each gene expression followed a standard normal distribution.

**Genotyping and imputation**. Genomic DNA was extracted from blood samples collected from 218 individuals. Genotyping was performed with Illumina Omni2.5 BeadChip array, with coverage of approximately 2.5 million markers. Variants with missing call rates >1%, MAF < 1%, or Hardy–Weinberg equilibrium (HWE) test $P$-value <$1 \times 10^{-6}$ were excluded, and individuals with missing call rates >1% were removed using Plink1.9. This produced an initial count of 1.4 million SNPs for 215 genotyped individuals. Of these, a total of 135 children also had gene expression data from cord blood (i.e. overlap with the 152 individuals described previously), and 106 and 119 individuals had gene expression data passing QC for both resting and stimulated conditions of monocyte/macrophage-enriched cultures and T-cell-enriched cultures, respectively, and 95 individuals had post-QC data for all four cultures.

We performed genotype imputation using the Michigan Imputation Server[60] with Haplotype Reference Consortium (HRC) release r1.1 as the reference panel. Phasing was performed using Eagle v2.3 and genotype imputation was performed using Minimac3 v1.0.4. After filtering out variants with low imputation accuracy ($R^2 < 0.3$), 12.7 million SNPs remained. For eQTL analysis, we focused on 4.3 million SNPs with MAFs ≥ 10%. The MAF cut-off used here was suggested by an eQTL simulation study in order to avoid inflated false positives in low-frequency variants given our limited sample size[61].

**Cis-eQTL mapping and conditional analysis**. To identify cis-eQTLs within each cell type and treatment group, we performed linear additive regression to model the effect of each SNP located within 1 Mb of the transcription start site (TSS) of the corresponding gene using the *Matrix eQTL* R package (v2.2)[62]. The sample size for eQTL mapping in each experimental condition was: 116 for resting myeloid cells, 125 for LPS-stimulated myeloid cells, 126 for resting T cells, and 127 for PHA-stimulated T cells. Genotype data were recoded as 0, 1, 2 based on the dosage of the HRC alternative allele. Gender, first three genotype PCs and first ten PEER[63] factors capturing technical variation in transcriptomes were included as covariates in the linear model.

We applied a hierarchical correction procedure to correct for multiple testing[61]. Firstly, nominal $P$-values for all cis-SNPs from *Matrix eQTL* were adjusted by multiplying the number of effective independent SNPs for each gene (local correction), which was estimated by *eigenMT* based on genotype correlation matrix[64]. Secondly, the minimum locally adjusted $P$-value for each gene was kept and the FDR of significant genes was controlled at 5% using the Benjamini–Hochberg (BH) FDR-controlling procedure (global correction)[65]. Genes with global FDR ≤ 0.05 were considered significant eGenes. Thirdly, to obtain the list of significant eSNPs for each eGene, the locally adjusted minimum $P$-value corresponding to the global FDR threshold of 0.05 was calculated, and SNPs with a locally adjusted $P$-value lower than the threshold were considered significant eSNPs.

Next, we performed conditional analyses to identify additional independent eQTL signals for each eGene. The gene-level $P$-value nominal thresholds calculated in the hierarchical multiple-testing correction (eigenMT-BH) were used to determine significant associations: the locally adjusted minimum $P$-value corresponding to the global FDR threshold of 0.05 multiplied by the number of estimated independent SNPs for each gene. We used a two-step conditional analysis scheme as follows:

*Forward stage*. For each eGene, the number of independent cis-eQTL signals was learnt from an iterative procedure. We started from the top SNP with the minimum $P$-value for the eGene, which was added as a covariate in the linear model to test for cis-eQTLs. If any significant SNPs (with $P$-values smaller than the gene's nominal threshold) were identified, the new top SNP identified in this iteration was added to the list of independent eQTL signals. In the next iteration of eQTL mapping, all previously identified eSNPs were adjusted for as covariates. The forward stage terminated if no additional significant associations were identified.

*Backward stage*. In this stage, the final list of significant SNPs representing each independent eQTL signal was determined. Let the list of independent SNPs for each eGene obtained from the forward stage be $SNP_1, SNP_2, SNP_3, \ldots, SNP_M$, where $M$ is the number of independent eQTL signals. Each of the independent eQTL signals was tested separately using a leave-out-one model adjusting for all other SNPs in the list as covariates. For example, when the $i$th eQTL signal was tested, $SNP_1, \ldots, SNP_{i-1}, SNP_{i+1}, \ldots, SNP_M$ were added as covariates together with other covariates used in the original eQTL scan. The final set of independent eQTLs comprised of the eSNPs that remained significant in the backward stage.

For genes that had eQTLs in more than one experimental condition, we also applied conditional analyses to identify independent eQTL signals between conditions. More specifically, to determine whether two eQTL signals for an eGene identified in two experimental conditions were independent or the same signal, we adjusted for the top eSNP in one condition by adding it as a covariate in the linear model and performed eQTL scan again in the other condition. If any SNP was significant in the conditional model using the $P$-value threshold determined by the hierarchical correction procedure (eigenMT-BH), we considered these two eQTLs as independent signals. If none were significant in the conditional model, we considered it as shared eQTL signal between two conditions or lack of power to detect the independent signal.

**Comparison of neonatal vs adult cis-eQTLs**. We downloaded full summary statistics of cis-eQTL analysis in adults from a response eQTL study in monocytes (Kim-Hellmuth et al. study[11]) and from the eQTL analysis in naïve T cells from the BLUEPRINT project[22]. Kim-Hellmuth et al. mapped cis-eQTLs in resting and LPS-stimulated CD14+ monocytes with two different durations of LPS: 90 min and 6 h, which were obtained from 134 adults aged from 18 to 35 years[11]. Chen et al. mapped cis-eQTLs in naïve CD4+ T cells from 169 adults (mean age of 55 years) from the BLUEPRINT project[22]. We could not find full summary statistics of any response eQTL analyses in stimulated T cells, so we only compared eQTLs in resting T cells.

To identify myeloid eQTLs that were specific to neonates, we analysed gene-SNP pairs (involving 10,749 genes) that had summary statistics available in both our study and the Kim-Hellmuth et al. study using the *mashr* (v0.2.21) R package[20]. Z scores were used as input. We used the most significant SNP for each gene as the strong set to learn data-driven covariance matrices, and randomly selected around nine tests for each gene as the random set to learn correlation structure among null tests and to fit the mash model. Posterior summaries were calculated for top eSNPs of each eGene using the fitted mash model, and calculated the local false sign rate (LFSR) for the most significant SNPs of 105 (out of 136) and 332 (out of 376) eGenes in resting and LPS-stimulated myeloid cells, respectively. Summary statistics for tests involving 7501 genes were available in both our study and the BLUEPRINT project, and we tested the most significant SNPs of 639 out of 971 eGenes identified in resting T cells. Neonate-specific eQTLs were defined as eQTLs that had LFDR < 0.05 in our dataset but not in cells from adults.

We have also estimated the sharing of cis-eQTLs between neonates and adults using the Storey and Tibshirani $q$-value approach[23]. $\pi_1$ statistic was calculated to quantify the replication rate using the *qvalue* (v2.16.0) R package. We used the *pi0est* function to estimate the proportion of tests that were truly null ($\pi_0$) among a list of $P$-values, given the assumption that $P$-values of truly null hypotheses should follow a uniform distribution. The proportion of true positive $\pi_1$ was calculated as $1 - \pi_0$. To assess the sharing of eQTLs between two datasets, we focused on significant eQTL associations identified in one dataset, and used the corresponding $P$-values of the same list of tests (SNP-gene pairs) from the second dataset as input of the *pi0est* function.

**Enrichment analysis**. We performed enrichment analyses using *GARFIELD* (version 2) to investigate the enrichment patterns of cis-eQTLs using predefined features such as genic annotations from ENCODE, GENCODE, and Roadmap Epigenomics project provided by this tool[21]. *GARFIELD* evaluates enrichment using generalised linear regression models that account for allele frequency, distance to the nearest gene TSS, and LD. LD correlation based on the UK10K dataset is also provided by the software. In each experimental condition, we used $P$-values for all SNPs tested in cis-eQTL analysis. If a SNP was tested for association with multiple genes, the smallest $P$-value was kept. Enrichment odds ratios were calculated at various eQTL significance thresholds: $1 \times 10^{-3}, 1 \times 10^{-4}, \ldots, 1 \times 10^{-7}$ (Supplementary Fig. 4), because *GARFIELD* accepts one single threshold.

**Response eQTL detection**. Response eQTLs (reQTLs) were identified in myeloid cells and T cells separately. For each cell type, we focused on top eSNPs of eGenes that were significant in either resting or stimulated conditions. For eGenes that were significant in both conditions and for which two top eSNPs were not in high LD ($r^2 < 0.8$), we tested both of the top eSNPs; on the other hand, if the two top eSNPs were in high LD, we tested the more significant one, to reduce tests on redundant SNPs. In myeloid cells, 417 interaction tests involving 398 eGenes were performed, and 1959 tests involving 1749 eGenes were performed in T cells. Gene expression data in two conditions were combined within each cell type, and the following linear mixed-effects model was tested for eGene–top eSNP pairs using the *lmer* function in the *lme4* R package (v1.1-18-1)[66]:

$$y_i \sim x_i + c_i + x_i \times c_i + x_i^1 + \ldots + x_i^{14} + x_i^1 \times c_i + \ldots + x_i^{14} \times c_i + (1|S_i). \quad (1)$$

where $y_i$ indicates the expression level of an eGene for the $i$th sample, $x_i$ the SNP allele dosage, $c_i$ the condition (resting: 0 and stimulated: 1) in which the gene expression was measured, $x_i^1, \ldots x_i^{14}$ the 14 covariates used in the original eQTL mapping (gender, three genotype PCs, and ten PEER factors), and $S_i$ the individual from which the $i$th sample was taken. The term $x_i \times c_i$ models the interaction between the genotype and the condition, and $(1|S_i)$ indicates the individual-specific random effect for this paired study design.

We applied permutations to estimate empirical $P$-values for the interaction term. In each permutation step, the condition variable was shuffled within each individual, and the same linear mixed model was tested to get the permuted statistics for the interaction term[12,67]. The permutation-determined $P$-value for each interaction test was calculated as $(s + 1)/(n + 1)$, where $n$ was the total number of permutations (1000) and $s$ was the number of cases where the permutated statistics were more significant than the original observed ones. We added 1 to both the numerator a the denominator to avoid underestimating permutation $P$-values. BH FDR-controlling procedure was applied to the permutation $P$-values and significant interactions were identified at 5% FDR.

**Trans-eQTL identification**. To detect *trans*-acting genetic regulation of gene expression in each condition, we tested for associations between SNPs and genes that were located on different chromosomes using the same linear model and covariates as in the *cis*-eQTL mapping. We tried the following different approaches to deal with the multiple testing:

*Genome-wide FDR correction.* BH FDR-controlling procedure was applied to nominal P-values from all *trans*-association tests, and significant *trans*-associations were identified at 5% FDR[6].

*Gene-level FDR correction.* For each gene, the *P*-value of the top SNP was multiplied by $1 \times 10^6$, which was the estimated number of independent SNPs across the genome (calculated as 0.05 divided by the commonly used genome-wide significance threshold of $5 \times 10^{-8}$). To control gene-level FDR, a BH FDR-controlling procedure was then applied to the minimum adjusted P-values for all genes.

*Gene-level Bonferroni correction.* Bonferroni correction was used to control the gene-level FDR, by using a significance *P*-value threshold of $3.8 \times 10^{-12}$ ($5 \times 10^{-8}/13{,}109$, where the denominator indicates the number of genes). The Bonferroni correction was extremely conservative because the tests (or genes) were not independent with each other.

In resting myeloid cells as well as in LPS-stimulated myeloid cells, one *trans*-eQTL signal was significant in all three methods. At 5% genome-wide FDR level, we observed 10 and 15 eGenes with significant *trans*-eQTLs (*trans*-eGenes) in resting and PHA-stimulated T cells, respectively, corresponding to a nominal *P*-value threshold of $1.9 \times 10^{-10}$ in resting T cells and $2.5 \times 10^{-10}$ in PHA-treated T cells. The number of significant *trans*-eGenes dropped, respectively, to six and eight by using the gene-level FDR correction (corresponding to a nominal *P*-value threshold of $5.3 \times 10^{-12}$ in both conditions), and to five and seven by using the gene-level Bonferroni correction. The limited power was the major issue given the sample size; thus, we used genome-wide FDR correction, the least conservative method, to determine significant *trans*-eQTLs used in the downstream analysis[6].

**Mediation analysis**. We hypothesised that *trans*-eQTLs regulated the expression of distant genes through *cis*-mediators, or local genes whose expression was regulated by the same *trans*-eQTLs. To test this hypothesis, we focused on the *trans*-eQTLs that were also associated with adjacent *cis*-eGenes, meaning that the *trans*-eQTLs were also *cis*-eQTLs. For each *trans*-eGene–*cis*-eGene pair, we tested the *trans*-eSNP with the smallest P-value as the exposure, a *cis*-eGene as the mediator, and a *trans*-eGene as the outcome (Fig. 3b). In total, we tested 14 mediation trios: one from resting myeloid cells, one from LPS-stimulated myeloid cells, nine from resting T cells, and three from PHA-stimulated T cells.

We performed mediation test for the 14 trios using the *mediation* R package (v4.4.6)[68]. The effect of the exposure on the mediator ($a$) was estimated in *cis*-eQTL mapping. The effect of the mediator on the outcome ($b$) adjusting for the exposure and the effect of the exposure on the outcome ($c\prime$) adjusting for the mediator were estimated in the following multiple regression:

$$y_i \sim x_i + x_i^{cis} + x_i^1 + \ldots + x_i^{14}. \qquad (2)$$

where $y_i$ indicates the value of the outcome (or the expression level of the *trans*-eGene) for the $i^{th}$ sample, $x_i$ the exposure (or the eSNP allele dosage), $x_i^{cis}$ the mediator (or the *cis*-eGene expression), and $x_i^1, \ldots x_i^{14}$ the 14 covariates used in eQTL mapping. The estimates of $b$ and $c\prime$ were beta coefficients for $x_i^{cis}$ and $x_i$, respectively. The direct effect of the exposure on the outcome was quantified as $c\prime$, the indirect effect of the exposure on the outcome through the mediator was quantified as $a \times b$, and the total effect was the sum of the previous two effects. Complete mediation occurs when the direct effect $c\prime$ is zero after controlling for the mediator, and partial mediation happens when the direct effect is different from zero. To identify significant mediation trios (the null hypothesis $H_0 : ab = 0$), we used a nonparametric bootstrap method (10,000 simulations) implemented in the *mediation* R package for variance estimation and P-value calculation. BH FDR-controlling procedure was applied to correct for multiple testing.

**Genetic overlap of eQTLs and diseases**. We downloaded publicly-available GWAS data for the following immune-mediated diseases: allergic disease (asthma, hay fever, or eczema), allergic rhinitis, allergic sensitisation, asthma (childhood-onset and adult-onset asthma), inflammatory bowel disease including its two subtypes—Crohn's disease and ulcerative colitis, celiac disease, autoimmune thyroid disease, juvenile idiopathic arthritis, multiple sclerosis, narcolepsy, primary biliary cirrhosis, primary sclerosing cholangitis, psoriasis, rheumatoid arthritis, systemic lupus erythematosus, and type 1 diabetes. We also analysed the summary statistics from a GWAS of educational attainment as a negative control. References of the GWAS studies are in Supplementary Information. These datasets contained summary statistics obtained using European populations for both significant and non-significant genetic variants, and GRCh37 genomic coordinates were available.

We performed enrichment tests for each of the four sets of significant eQTLs using *GARFIELD*[21]. Generalised linear models were applied to test for enrichment in eQTLs of variants associated with the above diseases at a significance threshold of $1 \times 10^{-6}$. Bonferroni correction was applied to correct for multiple testing,

where the number of tests was the number of GWAS datasets (24) multiplied by the number of eQTL datasets (4), and the Bonferroni-adjusted P-value threshold was $5.0 \times 10^{-4}$, adjusting for the $4 \times 25$ (100) tests.

**Colocalisation of cis-eQTLs with disease associations**. We applied a Bayesian method implemented in the *coloc* R package (v3.1)[25] to test whether any of the disease-associated GWAS loci shared the same causal variants with the *cis*-eQTLs. Full summary statistics were required to run the colocalisation analysis using *coloc*. For loci where *cis*-eQTLs were also associated with diseases at a *P*-value threshold of $1 \times 10^{-6}$, colocalisation test was performed on a 400-kb window centered on the top *cis*-eSNP. For each locus, colocalisation test was performed on overlapping SNPs where both eQTL and GWAS summary statistics were available. We excluded regions where not enough SNPs (<25) were available for colocalisation test. Selection of different prior probabilities of a SNP being causal for both of the traits affects the posterior support for colocalisation. To be conservative, we used a lower prior probability of $1 \times 10^{-6}$ instead of the default value of $1 \times 10^{-5}$.

For each locus, the Bayesian method assessed the support for the following five exclusive hypotheses: no causal variants for either of the two traits ($H_0$), a causal variant for gene expression only ($H_1$), a causal variant for disease risk only ($H_2$), distinct causal variants for two traits ($H_3$), and the same shared causal variant for both traits ($H_4$). The package estimated posterior probabilities ($PP_0$, $PP_1$, $PP_2$, $PP_3$, $PP_4$) to summarise the evidence for the above five hypotheses. High $PP_1$ or $PP_2$ and low $PP_3 + PP_4$ indicate a lack of power to identify the causal signals[25]. We excluded loci where $PP_3 + PP_4 < 0.8$, and focused on loci with strong evidence support for shared causal variants ($H_4$), i.e. ratio of $PP_4$ to $PP_3 \geq 5$.

To assess if our significant colocalisations were robust across a range of different p12 priors (from $1 \times 10^{-9}$ to $1 \times 10^{-5}$), we performed the sensitivity analysis using the *sensitivity* function in a newer version of the *coloc* R package (v4.0). All our significant colocalisations passed the sensitivity analysis, and $PP_4$ values using different priors can be found in Supplementary Data 4.

**Mendelian randomisation analysis**. To investigate causal effects of eGene expression on the above immune-mediated diseases, we performed a two-sample Mendelian randomisation (MR) analysis. Summary statistics from both our eQTL and external GWAS studies were required, including beta coefficient and its standard error, effective allele (based on which the beta was estimated), the other allele, and P-value. For each disease trait, we used *cis*-eSNPs that were also included in the GWAS dataset, and removed ambiguous variants (if any) using the *TwoSampleMR* R package (v0.4.14)[69]. We then selected LD pruned ($r^2 < 0.1$) *cis*-eSNPs as genetic instrumental variables (IVs). We focused on eGenes with at least three genetic IVs available, and performed the following MR methods implemented in the *MendelianRandomization* R package (v0.4.1)[70]: inverse variance weighted (IVW), weighted median, weighted mode, and MR Egger. These methods have different assumptions for valid IVs: IVW assumes that all IVs are valid; weighted median assumes that valid IVs contribute to more than 50% of the weight; weighted mode assumes that the largest group of IVs are valid; MR Egger regression, which is the least sensitive, assumes that the pleiotropic effects of IVs are not correlated with the genetic effects on exposure. We excluded the causal associations for which the intercept in the MR Egger method was significantly not equal to 0, indicating significant average pleiotropic effects. Gene expression was considered to have suggestive evidence of causal effects when at least three out of the four methods provided significant P-value (≤0.05). In the 'Results' section, we reported the statistics of the weighted mode method, which has the least assumption among all methods except the MR Egger, but is more sensitive than MR Egger.

**Reporting summary**. Further information on research design is available in the Nature Research Reporting Summary linked to this article.

## Data availability

The raw microarray data, normalised gene expression data, and full summary statistics of the eQTL and response eQTL analyses have been deposited in ArrayExpress with the accession code E-MTAB-8977. GWAS summary statistics used in this study are from the following publicly available sources: ImmunoBase (https://www.immunobase.org/), LD Hub (http://ldsc.broadinstitute.org/ldhub/), and GWAS Catalog (https://www.ebi.ac.uk/gwas/). The transcriptome profiles used to infer cell type abundances (CIBERSORTx) are available on https://cibersortx.stanford.edu. The GENCODE reference data are available at https://www.gencodegenes.org. The Haplotype Reference Consortium (HRC) r1.1 imputation reference panel was available at https://imputationserver.sph.umich.edu/.

## Code availability

The analysis scripts for eQTL mapping, colocalisation, and Mendelian Randomisation are available at https://github.com/QinqinHuang/CAS_eQTL.

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

## Acknowledgements

We thank the participants, their parents, and project staff of the Childhood Asthma Study. This research was supported by the NHMRC of Australia (project grant no. 1049539 to M.I. and K.E.H., Fellowships 1061409 to K.E.H., and 1061435 to M.I.). This research was supported by core funding from: the UK Medical Research Council (MR/L003120/1), the British Heart Foundation (RG/13/13/30194; RG/18/13/33946) and the National Institute for Health Research [Cambridge Biomedical Research Centre at the Cambridge University Hospitals NHS Foundation Trust]. This research was supported in part by the Victorian Government's Operational Infrastructure Support Program and the UK National Institute of Health Research. M.I. and S.C.R. are funded by the National Institute for Health Research [Cambridge Biomedical Research Centre at the Cambridge University Hospitals NHS Foundation Trust]. This work was supported by Health Data Research UK, which is funded by the UK Medical Research Council, Engineering and Physical Sciences Research Council, Economic and Social Research Council, Department of Health and Social Care (England), Chief Scientist Office of the Scottish Government Health and Social Care Directorates, Health and Social Care Research and Development Division (Welsh Government), Public Health Agency (Northern Ireland), British Heart Foundation and Wellcome. K.E.H. is supported by a Senior Medical Research Fellowship from the Viertel Foundation of Australia. The views expressed are those of the authors and not necessarily those of the NHS, the NIHR or the Department of Health and Social Care.

## Author contributions

M.I. conceived this project and supervised the work. K.E.H. and P.G.H. supervised the work. P.G.H. and P.D.S. are the principle investigators of the Childhood Asthma Study. Q.Q.H. carried out all analyses and wrote the paper. H.H.F.T. carried out quality control of the genotype data and Q.Q.H. carried out genotype imputation. M.I, H.H.F.T., and S.M.T. helped writing and editing the paper. H.H.F.T. contributed to the biological interpretation. H.H.F.T. and S.M.T. helped with quality control and data analysis. D.M. performed the stimulation experiments and RNA extraction. S.C.R. provided the codes for the Mendelian randomisation analysis and helped with the interpretation. A.P.N. helped with quality control of the microarray data, and the colocalisation analysis and its interpretation. M.B. was the project manager. A.S. contributed to the interaction model of the reQTL analysis. A.B. helped with quality control and normalisation of the microarray gene expression data. B.J.H. helped with lab work. C.C.K. contributed to the quantification of microarray gene expression data. All authors approved the paper.

## Competing interests

The authors declare no competing interests.
