## [Peer Review File · Nature Communications]

Reviewers' comments:

Reviewer #1 (Remarks to the Author):

This paper describes a comprehensive investigation into the genetics of gene expression in resting and stimulated monocytes and CD4+ T cells from cord blood. The authors identified numerous cis and trans eQTL and showed how some cis regulatory factors mediate trans effects. They then investigated genetic overlap with autoimmune and allergic diseases and used MR to identify potential causal effects of neonatal gene transcription on later disease risk. The authors are to be congratulated on a thorough and well-performed study. The paper is generally clear and well written – the first figure on study design is a particularly helpful overview of the complex project. The summary statistics of this study would provide an extremely valuable resource to the research community. I have the following comments and queries.

1. The cord blood used in this study was collected at birth and assumed to be indicative of the perinatal period, but cord blood may also to an extent reflect in utero exposures. Such in utero exposures may be influenced by the maternal genotype, which is correlated with the child's genotype. Can the authors comment on the possible implications of this for the results of their study, which has looked at only the child's genotype?
2. There is clear evidence from the MR analyses that the gene expression in the early-life immune cells studied is potentially causal to the risks of certain diseases, but can the authors comment on the extent to which this is specific to early life exposures and expression? Can the comparisons with expression studies in adult cells inform as to whether expression in early vs. later life is more important? It would be helpful to consider this in the discussion.
3. The paper shows good evidence for genetic overlap between neonatal gene expression in the cells tested and immune-mediated disease. One would expect that there is more overlap/enrichment for immune-mediated diseases than for non-immune-mediated diseases and traits. Have the authors checked this?
4. Introduction: "eQTLs identified...are enriched for genetic variants associated with development (e.g. adult height)". It is not clear to me how adult height is a trait indicative specifically of "development". Do the authors mean "growth" or "anthropometric traits"?
5. A few things would improve the presentation of figure 2 for the reader:
 - a. Panel A legend says percentages of significant results are in green, but it looks blue to me!
 - b. Throughout the figure, the results are referred to as "significant" or "non-significant", but it is not clear what the criteria for significance are. Much more informative to put the p value threshold or FDR threshold there.
 - c. In the legend to panel C, the effect size is referred to as a "change in gene expression per allele". I think it would be more helpful to refer to this as "difference in gene expression..." (if I have understood correctly) because "change" could be initially interpreted as change upon stimulation.

Reviewer #2 (Remarks to the Author):

Huang et al. have analyzed a subset of the Childhood Asthma Study to identify eQTLs and response eQTLs (reQTLs) in monocytes and T cells isolated from resting and stimulated PBMCs (n=116-127). They apply cis- and trans-eQTL mapping, colocalization and MR tools, however, major (and minor) technical problems would need to be solved before being able to interpret results and draw meaningful conclusions from this study. Besides the analytical problems, questions such as how do neonatal (r)eQTLs differ from adult (r)eQTLs? and what is the information gained by studying neonatal (r)eQTLs? remain unanswered. This work currently provides few novel discoveries and the relatively small sample size limits their power to do truly outstanding comprehensive analyses.

Here is a list of comments on specific points:

1) Methods do not meet current standards:

- Substandard cell isolation protocol: Monocytes were isolated using an adherence protocol instead of MACS isolation or flow cytometry sorting, which are superior isolation protocols. The adherence protocol is a quick and easy procedure, but it can induce cell activation (Fuhlbrigge et al., 1987; Haskill et al., 1988) and purity is in general lower and more variable. Have the authors characterized the isolated cell populations (both monocytes and T cells) in terms of surface marker expression to quantify purity and cell activation status? If so, this information should be added and more importantly included as covariates in the eQTL model.
- Cell stimulation in mixed PBMC instead of purified cell types: Neither the main text nor Fig.1 clearly indicate that monocytes and T cells were isolated after (not before) immune stimulation. This is very uncommon and clearly deviates from all previous immune response eQTL studies. By stimulating a mixed cell population such as PBMCs instead of purified cell types you lump together both direct (e.g. TLR4 and TCR-mediated) as well as indirect (e.g. cell-cell interaction, cytokine effects from other cell types, etc.) immune response signatures. Not only does it rule out the mechanistic localization of eQTL effects but it also impedes the comparison with existing immune eQTL studies. If this protocol was chosen on purpose the authors should provide rationale and literature to backup this decision.
- Gene expression was quantified via microarray technology, which was replaced by RNA sequencing technology a long time ago and does not reflect current gold standard anymore.
- Dated eQTL analysis: Permutation based eQTL analysis as implemented in fastQTL or qtltools are gold standards since 2015 and should be used instead of MatrxieQTL + eigenMT correction. Genomewide significant eQTL results might be similar but it's important to map these standard cis-eQTLs properly.
- eQTL sharing analysis shown in Fig. 2B is not properly done and the resulting 74% condition-specificity is way too high. Condition-specificity cannot be simply defined as reaching an arbitrary (in this case the eigenMT-BH) p-value threshold in both conditions or not. Statistical frameworks implemented in tools such as eqtlBMA/mashr or Metasoft/Metatissue are well suited to quantify eQTL specificity and I highly recommend redoing the analysis.
- The eQTL replication test (Table S6) is not properly done lacking the inclusion of the effect sign besides the p-value of the replication data sets. I strongly recommend though using $n1$ statistics (Storey et al.) instead, which is more common if not standard. Full summary statistics are needed for this analysis but these are mostly/easily available nowadays (e.g. Monocytes - Kim-Hellmuth et al., T-cells - Ye et al.).
- reQTL mapping: while the interaction model is reasonable the corresponding permutation scheme is not valid as the bond between genotype and expression still remains (see Buzkova et al. Annals of Human Genetics 2010, and Buzkova, Epidemiol. Methods 2016 for explanation and solutions). Did the authors check if p-values are calibrated? Permutation schemes for interaction terms are non-trivial (e.g. parametric bootstrap) and one could consider applying a very conservative correction instead (Bonferroni or more stringent). With the current permutation scheme the proportion of reQTLs in Fig.2A are highly overestimated and basically useless in the current form. Also, Table S7+8 do not provide the lmer nominal pvalues (only permutation p is listed).
- trans-eQTL analysis is still challenging and is not recommended for studies with low sample size. Without evidence of replication it is difficult to assess how valid the findings of this study are.
- colocalization analysis: coloc is very sensitive to the prior selection. Modifying just one prior often doesn't solve the problem. Did the authors check if posterior probabilities of their loci stay high across a range of different $p1$, $p2$ and $p12$ priors? I would recommend running this analysis (e.g. applying bayesian model averaging) or optionally using more advanced colocalization methods such as eCAVIAR or enloc.

2) Lack of novelty and impact for the research community:

- None of the analyses try to answer the question what we can learn from neonatal samples that we miss in adult samples. Did the authors check if there are any differentially expressed immune response genes (DEG) between neonatal vs adult samples? The number of DEG in a small pilot subset is a good proxy for the number of (r)eQTLs expected to differ between two conditions on a population scale. If there are hardly any differences in immune response between newborns vs

adults to begin with it is extremely unlikely that an eQTL study in neonates is that much more informative than a study in adults.

- To compare neonatal vs adult eQTLs one could use methods such as mashr that are based on summary statistics. Given the differences in stimulation/isolation protocol the comparison of stimulated cells or reQTLs might be limited but comparing baseline (=resting) conditions might be a good start.

- eQTLs with opposing effects in different cell types/conditions/tissues have been shown to be mostly artefacts due to "LD contamination". This happens when the studied variant is actually tagging an additional or independent eQTL (with opposite direction) that is only active in one of the conditions. It then looks like the direction of the eQTL changes between conditions but you are actually capturing two different eQTLs. This might well be the case for DDT and ZNF589 shown in this study.

- the vignette shown in Fig. 3D at the RPS26 locus has been described before in T cells (e.g. Kasela et al. 2017) and as the authors indicate its mediation role has been shown in GTEx already and is therefore not a novel finding.

3) Specific questions:

- it is not clear to me why the number of eGenes in monocytes are so much smaller than in T cells. Previous studies (Fairfax et al. 2012, Peters et al. 2016) that compared monocytes vs other cell types showed the opposite (more eGenes in monocytes than in other cell types). The authors should try to find out if any technical/analytical factors led to this discrepancy. The number of expressed genes shown in Fig. S1 are suspiciously low (they were not all on the same e.g. low performing chips, right?) and the distribution might reflect the heterogeneity of the monocyte population. Please provide literature and any additional QC plots to back up the validity of the monocyte expression data set.

- Fig. 4A: instead of just color-coding the cell type it would be more informative to show the actual PP4 for all gene-trait pairs as a color gradient in the heatmap. Supplementary figures or tables showing the PP4s of the other 3 conditions would be helpful to assess how cell type- or condition-specific these colocalizations are.

- Please indicate if any of the vignettes have been previously found in bigger eQTL data sets.

4) Data availability:

Please clarify if and in which form data will be available for the research community. Ideally multiple layers of data (raw, processed, eQTL summary statistics) should be made available.

5) Minor points:

- Fig. 1: The study design cartoon should clearly indicate that PBMCs (not monocyte and T cells) were stimulated. Additional arrows between the culture plates and the Monocyte/T cells label will help to clarify that the monocyte and T cell isolation was done after the stimulation. It might also be more correct to show only three cartoon plates instead of four as (I assume) resting adherent and non-adherent cell subsets were isolated from the same sample.

- Fig. 3:

- given the small number of trans-eQTLs, results might be easier to read in a simple table instead of a circosplot.

- are microRNAs quantifiable in these samples or were small RNAs removed (poly-dT Primer, RNA size selection) before cDNA synthesis? This information is missing in the corresponding methods section.

- Fig. S2: please provide rationale for changing the eQTL significance level to 1×10^{-5} . Are results stable at different significance thresholds?

- for all suppl. Figures with colocalization regional plots (Fig. S5/6/7/10/11) I would recommend plotting only variants that are present both in the eQTL as well as in the GWAS traits, which reflect the set of variants (=intersect) that coloc uses. Currently the eQTL plots have many more variants and some GWAS traits seem to have very few data points (see Fig. S10).

- Fig. S9: I might have missed it but why is the IBD/PHA T plot missing?

- How do the HLA-C colocalization regional plots look like?

- Methods: concentrations of LPS and PHA used to stimulate cells are missing

- given above comments, the title "reveal the origins of autoimmune and allergic disease risk"

might be a bit too ambitious.

Reviewer #1 (Remarks to the Author):

1. The cord blood used in this study was collected at birth and assumed to be indicative of the perinatal period, but cord blood may also to an extent reflect in utero exposures. Such in utero exposures may be influenced by the maternal genotype, which is correlated with the child's genotype. Can the authors comment on the possible implications of this for the results of their study, which has looked at only the child's genotype?

Thank you for raising this point. We have updated the discussion section in the manuscript with the following:

“Our study had limitations. Cord blood may be under the influence of in utero exposures (e.g. smoking, drug exposures, maternal stress) which may confound associations with maternal, placental or foetal genetics¹, and may affect neonatal gene expression through epigenetic mechanisms²⁻⁴. In utero exposures, epigenetics, or maternal genetics were not measured in this cohort.”

To our knowledge, while there are many studies exploring the direct effect of *in utero* exposures on infant and transgenerational health, there are currently few studies that directly link interaction between *in utero* exposures and maternal genetics to infant gene expression. This is certainly an appealing target for further research. Nonetheless, complex traits such as susceptibility to environmental factors often interact with human genetics on a polygenic scale⁵⁻⁷, with variants distributed across the entire genome each having a small effect. Our study was more focussed on infant *cis* eQTLs, genetic variants with strong effects located near their target gene. Thus, although potential confounding by *in utero* exposures may reduce power to detect *cis*-eQTL signals, it is unlikely to create spurious eQTL associations.

2. There is clear evidence from the MR analyses that the gene expression in the early-life immune cells studied is potentially causal to the risks of certain diseases, but can the authors comment on the extent to which this is specific to early life exposures and expression? Can the comparisons with expression studies in adult cells inform as to whether expression in early vs. later life is more important? It would be helpful to consider this in the discussion.

To check whether genes with causal effects in neonates were also relevant to adult tissues, we ran MR analyses for the relevant genes in adults with adult eQTLs genetic instruments. We used monocyte eQTLs (resting and stimulated conditions) from Kim-Hellmuth *et al.*⁸ and T-cell eQTLs from BLUEPRINT⁹. For our analysis, we excluded three monocyte genes (*HLA-C*, *HLA-DQA1*, and *HLA-DQA2*) not tested in Kim-Hellmuth *et al.*, and eight resting T cell genes (*ACER3*, *BTN3A2*, *FDFT1*, *FN3KRP*, *HLA-C*, *HLA-DQA2*, *MGST3*, *SERPINB6*) not tested in BLUEPRINT. Four genes had eQTLs shared between neonatal and adult cells identified by a multivariate adaptive shrinkage (mash) model (updated **Methods** section): *HLA-DRB5* and *THBS1* in LPS-stimulated monocytes, and *ERMP1* and *MICB* in resting T cells. However, these had at most two independent instruments each, and so were insufficient for MR analyses (which require ≥ 3). Meanwhile, we found that the eQTL of *TSPO* in resting T cells was specific to neonates in our study.

Ultimately, it is difficult to judge whether infant or adult gene expression is more important for these particular genes, given that we are dealing with different human and cell populations for each subset of data (neonatal expression, adult expression, genome-wide associations with diseases). All we can conclude for now is that some genes may act perinatally to affect risk of adult disease (e.g. *TSPO* as above), while others may only be physiologically-important in adulthood. There is at least some evidence that certain eQTLs were shared between or specific within neonates and adults, irrespective of causal link to disease (see mash model in **Methods** and, and response to Comment 7).

3. The paper shows good evidence for genetic overlap between neonatal gene expression in the cells tested and immune-mediated disease. One would expect that there is more overlap/enrichment for immune-mediated diseases than for non-immune-mediated diseases and traits. Have the authors checked this?

We did indeed find that overlap was greater for immune-mediated disease than non-immune-mediated traits. We have updated the **Results** section with the following:

*“First, we performed GARFIELD enrichment analyses to test for significant overlaps between the cis-eQTLs and variants associated with immune-mediated disease in genome-wide association studies (GWAS; **Methods**). We found widespread enrichment amongst cis-eQTLs for genetic variants associated with diseases such as allergic disease (asthma, hay fever, or eczema) and inflammatory bowel disease. Conversely, same analysis with non-immune-related traits such as educational attainment¹⁰ identified limited enrichment (**Figure S5**). Specifically, we did not observe significant enrichment of GWAS signals for educational attainment in monocyte eQTLs; there was evidence of enrichment in resting and PHA-stimulated T cells but enrichment was not as strong (effect sizes 0.43 and 0.52 respectively) compared to immune-mediated diseases (ranging from 1.13 to 7.37).”*

4. Introduction: “eQTLs identified...are enriched for genetic variants associated with development (e.g. adult height)”. It is not clear to me how adult height is a trait indicative specifically of “development”. Do the authors mean “growth” or “anthropometric traits”?

Yes, we mean “foetal growth”. We have updated this in the manuscript.

5. A few things would improve the presentation of figure 2 for the reader:

a. Panel A legend says percentages of significant results are in green, but it looks blue to me!

We have updated the figure legend.

b. Throughout the figure, the results are referred to as “significant” or “non-significant”, but it is not clear what the criteria for significance are. Much more informative to put the p value threshold or FDR threshold there.

We used a hierarchical multiple testing procedure (**Methods**) to determine significant eQTL association, which involves a local correction (adjusting for multiple tests with cis-SNPs for each gene) and a global FDR correction (adjusting for multiple genes). Each gene has a nominal p-value threshold, and these p-value thresholds are usually slightly different. We have updated the legend to show that the “significant” signals are at a 5% false discovery rate (global correction).

c. In the legend to panel C, the effect size is referred to as a “change in gene expression per allele”. I think it would be more helpful to refer to this as “difference in gene expression...” (if I have understood correctly) because “change” could be initially interpreted as change upon stimulation.

We thank the reviewer for raising this point, and have updated the legend to panel C to make it less confusing.

Reviewer #2 (Remarks to the Author):

1. Substandard cell isolation protocol: Monocytes were isolated using an adherence protocol instead of MACS isolation or flow cytometry sorting, which are superior isolation protocols. The adherence protocol is a quick and easy procedure, but it can induce cell activation (Fuhlbrigge et al., 1987; Haskill et al., 1988) and purity is in general lower and more variable.

The experiments reported in this paper are part of a birth cohort study in which neonatal blood cell volumes collected were small (due to conditions imposed by ethics committee), mandating the use of micro methods for *in vivo* studies. MACS isolation and flow cytometry require much higher cell numbers than were available to us, necessitating the use of adherence purification for this phase of the study. We do not agree that the adherence protocol is ‘substandard’, as the isolation of adherent cells to produce enriched monocytes has been a well-established protocol in cellular immunology for more than 30 years. Analogous to what occurs *in vivo*, adherence to substratum *in vitro* is acknowledged to induce low level activation of monocytes, but why this should be considered a shortcoming in the present study is unclear. In order to carry out this study, it was necessary to first activate the monocytes, and all standard monocyte activation protocols are carried out under conditions that allow monocyte adherence to occur, because deliberately preventing adherence (i.e. maintaining them in suspension) results in suboptimal levels of activation. Both MACS isolation and

flow cytometric cell sorting also induce varying levels of cell activation as a result of engagement of cell surface molecules with the antibodies employed to identify target cells, shearing forces encountered in the cytometer, and/or active adherence to MACS beads. There is no valid evidence to show that adherence-associated activation differs either quantitatively or qualitatively from that obtained via sorting or selection employing MACS beads (or their equivalent, DYNA beads used in our study to isolate the CD4⁺ T cells)¹¹.

2. Have the authors characterized the isolated cell populations (both monocytes and T cells) in terms of surface marker expression to quantify purity and cell activation status? If so, this information should be added and more importantly included as covariates in the eQTL model.

While this information was not collected at the time of experimentation, we do not expect inclusion of these as covariates to significantly change our eQTL results. The PEER factors in the eQTL model correct for latent variation^{9,12,13}, and we would expect that they would capture these differences. As far as we are aware, no previous response-eQTL study involving purified immune cells has included such information as covariates in their eQTL model (e.g. Kim-Hellmuth *et al.*⁸ and Ye *et al.*¹⁴ as Reviewer 2 suggested in point 7).

3. Cell stimulation in mixed PBMC instead of purified cell types: Neither the main text nor Fig.1 clearly indicate that monocytes and T cells were isolated after (not before) immune stimulation. This is very uncommon and clearly deviates from all previous immune response eQTL studies. By stimulating a mixed cell population such as PBMCs instead of purified cell types you lump together both direct (e.g. TLR4 and TCR-mediated) as well as indirect (e.g. cell-cell interaction, cytokine effects from other cell types, etc.) immune response signatures. Not only does it rule out the mechanistic localization of eQTL effects but it also impedes the comparison with existing immune eQTL studies. If this protocol was chosen on purpose the authors should provide rationale and literature to backup this decision.

The stimulation of unfractionated PBMCs prior to the isolation of specific immune cells was deliberately carried out in our study to approximate *in vivo* conditions – in which virtually all stimulation events occur within mixed cell environments within which interactions between myeloid and lymphoid cells are free to occur. Stimulation of isolated cells will not represent *in vivo* conditions. Importantly, the polyclonal T-cell mitogen PHA employed here was deliberately chosen because (analogous to natural environmental antigens) it is “accessory cell dependent”, i.e. PHA-mediated T-cell activation (as is the case with antigen-mediated activation) requires costimulatory signals delivered to the T cells, usually by myeloid cells.

We have updated the **Methods** section to emphasise this. We have also updated Figure 1 to clarify that stimulation was carried out in mixed PBMC cultures.

4. Gene expression was quantified via microarray technology, which was replaced by RNA sequencing technology a long time ago and does not reflect current gold standard anymore.

We would note that there is no gold standard for gene expression analysis as choice of technology depends on sample limitations, cost, speed, analytic options, and the biological question of interest. We considered both RNA sequencing and microarrays when designing the current study. The choice of microarray technology allows for correction of technical variation and proper normalisation per-chip and per-gene. Studies have shown that the technical variation in microarray analysis can be sufficiently controlled, and the biological variation is the greatest source of variation¹⁵⁻¹⁷. There are mature pipelines to deal with the technical issues with microarray data. RNA sequencing technology, while comprehensive, does not constitute a gold standard. It is well-known to be prone to biases from various sources such as 3' bias¹⁸ and PCR duplication bias^{19,20}. Constructing large-scale libraries with low input RNA, as is necessary in paediatric studies, would exacerbate these artefacts and potential introduce new ones^{21,22}.

Notably, gene expression microarrays are currently used in clinical diagnostics and clinical decisions. In comments below, Reviewer 2 also recommends that we compare our neonatal eQTLs with two adult eQTL datasets to demonstrate replication, yet these two studies (Kim-Hellmuth *et al.*⁸ and Ye *et al.*¹⁴) also used microarray technology. While we understand the desire to utilise the latest technologies, the limitations of our paediatric cohort combined with the need to minimise technical

effects meant that we had to make a conservative decision and select the most tried and robust technology available in this case.

5. Dated eQTL analysis: Permutation based eQTL analysis as implemented in *fastQTL* or *qtltools* are gold standards since 2015 and should be used instead of *MatrixeQTL* + *eigenMT* correction. Genomewide significant eQTL results might be similar but it's important to map these standard cis-eQTLs properly.

We appreciate the reviewer's recommendation; however, there is in fact no gold standard for eQTL analysis. Several approaches are of equivalent sensitivity and specificity, and we would argue that our method is equivalent in validity to the methods suggested by Reviewer 2. As we explained in the **Methods** section, we used a hierarchical correction procedure to adjust for multiple testing (which was extensively compared in Huang *et al*, *Nuc Acids Research* 2018²³). We did not use the FDR in the output of *MatrixeQTL* in our multiple testing correction. We used *MatrixeQTL* to perform linear regression, which is the same with that in *fastQTL* or *qtltools* (the latter is a toolset which includes the former). We have performed extensive simulation studies to explore the study design and analysis choices in cis-eQTL studies, and we found that hierarchical correction procedures showed calibrated FDR, and the differences amongst different hierarchical procedures are small²³. In our eQTL analysis, we used *eigenMT* to correct for multiple tests involving cis SNPs for each gene (local correction), and the BH-FDR controlling procedure to correct for multiple genes tested (global correction). *fastQTL* uses permutations in the local correction step, and our simulation analysis showed that the differences caused by different local correction methods were minor.

6. eQTL sharing analysis shown in Fig. 2B is not properly done and the resulting 74% condition-specificity is way too high. Condition-specificity cannot be simply defined as reaching an arbitrary (in this case the *eigenMT*-BH) p-value threshold in both conditions or not. Statistical frameworks implemented in tools such as *eQTLBMA*/*mashr* or *Metasoft*/*Metatissue* are well suited to quantify eQTL specificity and I highly recommend redoing the analysis.

We again appreciate the reviewer's recommendation, but we would note that our method of eQTL sharing analysis remains statistically rigorous. We conditioned on the most significant eQTL SNP in one condition, and applied the eQTL scan in the second condition; and we considered it as a shared eQTL if the eQTL signal in the second condition disappeared. We had to use p-value thresholds to determine if a variant was significant or not in the conditional analysis, thus thresholds from the original eQTL scan were utilised to be consistent in determining eQTLs.

That said, we also performed eQTL comparisons across four conditions using the *mash* model suggested by Reviewer 2²⁴. Condition-specific eQTLs were defined as eQTLs that were active in one condition with LFSR <0.05, and were not active in the other three conditions. We observed fewer condition-specific signals using *mash* model: top eSNPs of 2, 38, 208, and 419 eGenes were condition-specific eQTLs in resting monocytes, stimulated monocytes, resting T cells, and stimulated T cells, respectively, while our conditional analyses identified 24, 151, 504, and 887 condition-specific eQTLs. All of the condition-specific signals identified in monocytes using *mashr* were replicated in our conditional analysis, and 204 and 413 signals in resting and stimulated T cells, respectively, were identified using both methods. However, it is not possible to know definitively whether this is because *mash* has lesser power or conditional analysis has a higher false discovery rate. To investigate this further, we looked at several factors. Firstly, the hierarchical multiple testing correction method used in our study to determine significant eQTLs was more stringent than the *mash* model (*mashr* does not account for multiple testing), and there were many cases where genes had active eQTL signals determined by *mashr*, but the same genes did not have any significant eQTLs using our multiple testing correction. Secondly, the *mash* model focused on top SNPs per gene, thus it missed secondary eQTLs for genes with multiple independent eQTLs. Lastly, the *mash* model does not distinguish between causal eQTLs from those that are in LD with the causal variants, while this was better accounted for in the conditional analysis.

7. The eQTL replication test (Table S6) is not properly done lacking the inclusion of the effect sign besides the p-value of the replication data sets. I strongly recommend though using π_1 statistics (Storey *et al.*) instead, which is more common if not standard. Full summary statistics are needed for this analysis but these are mostly/easily available nowadays (e.g. Monocytes - Kim-Hellmuth *et al.*, T-cells - Ye *et al.*).

We would note that the π_1 statistic recommended by Reviewer 2 also does not account for the effect sign; it takes only p-values into account. To evaluate the replication rate, the Storey *et al.* method takes the p-values estimated in dataset 2 for the exact list of significant signals identified in dataset 1, and compares this list of p-values against the null distribution. We do not think it is straightforward to interpret this replication estimate. Furthermore, the π_1 statistic does not account for LD patterns, and the estimate is affected by the number of significant variants at each locus (it would be biased towards loci with more genotyped and imputed variants available). Besides, the π_1 statistic gives a single value indicating replication rate, from which we cannot determine which specific signals were replicated. It is more useful in specific circumstances; for example, when there are dozens of different cell types or conditions, it can tell us which cells or tissue types share more eQTL signals than others.

We applied the mash model to compare eQTLs identified in neonatal immune cells and that in adults (**Methods; Figure 2**). For monocytes, we used the eQTL summary data from Kim-Hellmuth *et al.*, which identified eQTLs in resting and LPS-stimulated (90min and 6 hours) monocytes from 134 adults⁸. In total 10,749 genes were tested in both our study and the Kim-Hellmuth *et al.* study. Summary data for 105 (out of 136) and 332 (out of 376) significant eGenes in our resting and stimulated monocytes, respectively, were also available in the Kim-Hellmuth *et al.* study. In resting monocytes, 7 (6.7%) eGenes were specific to neonates at LFSR <0.05, and the rest showed evidence of being active in both neonates and adults. For stimulated monocytes, 24 (7.2%) were specific to neonates, which is as expected given that the neonatal immune system shows distinct features compared with adult immune systems. For the comparison of T-cell eQTLs, we could not find the full eQTL summary data from the Ye *et al.* study suggested by reviewer 2. The stimulant used in the Ye *et al.* study to activate T cells was also different from what was used in our study, so we performed the comparison in resting T cells using the eQTL dataset from the BLUEPRINT project⁹. Summary data of tests for 7,501 genes were available in both our study and the BLUEPRINT project, and 639 out of 971 significant eGenes in resting T cells were tested in BLUEPRINT. At LFSR <0.05, 158 (24.7%) eGenes were specific to neonates.

It is worth noting that this analysis might underestimate the proportion of neonate-specific eQTLs, because we restricted our analysis to gene-SNP pairs that were available in both neonatal and adult datasets. Genes that were not expressed in the adult immune cells were excluded from the analysis, and eQTLs of these genes are likely to be specific to neonates. Also, mash does not distinguish between causal variants from those that are in LD with them, and it focuses on the top SNPs and does not take multiple independent signals into account.

8. reQTL mapping: while the interaction model is reasonable the corresponding permutation scheme is not valid as the bond between genotype and expression still remains (see Buzkova et al. Annals of Human Genetics 2010, and Buzkova, Epidemiol. Methods 2016 for explanation and solutions). Did the authors check if p-values are calibrated? Permutation schemes for interaction terms are non-trivial (e.g. parametric bootstrap) and one could consider applying a very conservative correction instead (Bonferroni or more stringent). With the current permutation scheme the proportion of reQTLs in Fig.2A are highly overestimated and basically useless in the current form. Also, Table S7+8 do not provide the lmer nominal p-values (only permutation p is listed).

We would acknowledge that the rationale for the permutation testing of response eQTLs could have been clearer. In contrast to regular eQTL mapping (no interactions), it is in fact desirable to maintain the bond between genotype and expression during permutation when detecting response eQTLs^{25,26}. In the reQTL mapping, we tested if the changes in eQTL effects across conditions were significant, given that genotype had effects on gene expression. Also, the papers and the parametric bootstrap method mentioned by reviewer 2 are actually trying to retain the genotype-expression correlation (or, the “main effects” in the papers).

We used the same method with the Alasoo *et al.* study²⁵. Permutations were performed to estimate empirical p-values (not to correct for multiple testing), and we applied BH FDR on those empirical p-values. We do not believe that the proportion of reQTLs was overestimated.

9. trans-eQTL analysis is still challenging and is not recommended for studies with low sample size. Without evidence of replication it is difficult to assess how valid the findings of this study are.

Indeed, our study is not well-powered for *trans*-eQTL detection. However, the *trans*-eQTLs are still informative as long as multiple testing is properly controlled. We applied the genome-wide FDR method used in the GTEx project, which also mapped *trans*-eQTLs in tissues with sample sizes ranging from 79 to 361¹³.

Reviewer 2 commented that one of our *trans*-eQTL on the *RPS26* locus was not a novel finding because previous studies have already reported this locus. We counter that it is evidence of replication and method validity, and remains scientifically-valuable.

10. colocalization analysis: coloc is very sensitive to the prior selection. Modifying just one prior often doesn't solve the problem. Did the authors check if posterior probabilities of their loci stay high across a range of different p1, p2 and p12 priors? I would recommend running this analysis (e.g. applying bayesian model averaging) or optionally using more advanced colocalization methods such as eCAVIAR or enloc.

We agree that *coloc* is sensitive to the prior selection; that is why we used a more stringent prior to be conservative. We have also performed sensitivity analysis using the *sensitivity()* function that has been implemented in the new version of *coloc*, which evaluates if the colocalisation is robust across a range of different p12 prior values ranging from 1×10^{-9} to 1×10^{-5} . All our significant colocalisations passed the sensitivity analysis. We also varied p1 and p2 priors from 1×10^{-4} to 1×10^{-5} , and the posterior probabilities of colocalisations stayed high.

2) Lack of novelty and impact for the research community:

To clarify, our study offers the several novel and impactful discoveries:

- (1) We identified novel eQTL signals that were not observed in adult immune cells, which can help us better understand the genetic background of the differences between neonatal and adult immune responses.
- (2) We dissected the mechanisms of several *trans*-acting eQTLs, which regulate gene expression of distant genes through *cis*-mediators.
- (3) We observed colocalisation of neonatal eQTLs and variants associated with immune-mediated diseases that develop in adults, and many of the colocalisations were cell type- or condition-specific. This aids our understanding of the regulatory role of disease-associated variants, and more importantly, the specific cellular context in which those variants exert their functional effects.
- (4) We identified genes with causal neonatal effects on risk of immune-mediated diseases, linking early-life gene expression profiles to later disease risk.
- (5) This is, to our knowledge, the first response eQTL study in neonatal immune cells, and the data generated in our study can serve as a useful resource for future studies.

11. None of the analyses try to answer the question what we can learn from neonatal samples that we miss in adult samples. Did the authors check if there are any differentially expressed immune response genes (DEG) between neonatal vs adult samples? The number of DEG in a small pilot subset is a good proxy for the number of (r)eQTLs expected to differ between two conditions on a population scale. If there are hardly any differences in immune response between newborns vs adults to begin with it is extremely unlikely that an eQTL study in neonates is that much more informative than a study in adults.

It has been well established that there are major differences between neonatal and adult immune systems, both in terms of immune cell gene expression²⁷, and at the macroscopic scale. Compared to adults, neonates are more susceptible to infectious diseases, partly because of reduced number and function of immune cells²⁸. There is a lack of immunological memory in neonates^{29,30}. In addition, the foetal and neonatal immune systems show an imbalance between TH1 and TH2 cell populations – foetal cytokine production is naturally biased towards TH2 cytokines, avoiding pro-inflammatory/TH1-cell-associated immune responses, which have been associated with spontaneous abortions²⁸⁻³². Ultimately, there is a need to determine whether subtle differences in neonatal immunity can influence adult disease in ways not detectable by examining adult immunity alone.

12. To compare neonatal vs adult eQTLs one could use methods such as mashr that are based on summary statistics. Given the differences in stimulation/isolation protocol the comparison of

stimulated cells or reQTLs might be limited but comparing baseline (=resting) conditions might be a good start.

We have taken this suggestion on board and compared neonatal vs adult eQTLs using *mashr* (see above responses to the comment on eQTL replication test).

13. eQTLs with opposing effects in different cell types/conditions/tissues have been shown to be mostly artefacts due to “LD contamination”. This happens when the studied variant is actually tagging an additional or independent eQTL (with opposite direction) that is only active in one of the conditions. It then looks like the direction of the eQTL changes between conditions but you are actually capturing two different eQTLs. This might well be the case for DDT and ZNF589 shown in this study.

We thank the reviewer for raising this point. We have updated the discussion section.

14. the vignette shown in Fig. 3D at the RPS26 locus has been described before in T cells (e.g. Kasela et al. 2017) and as the authors indicate its mediation role has been shown in GTEx already and is therefore not a novel finding.

The purpose of presenting the RPS26 locus as one example of our *trans*-eQTL findings was to demonstrate the informativeness and validity of the mediation analysis, despite our limited power (re: Reviewer 2’s previous comments on *trans*-eQTL analysis in point 9). We further note that Fig. 3C and 3D showed two *trans*-eQTLs for which we found strong evidence of *cis* mediation, and we identified various other *trans*-eQTL loci (Fig. 3A).

3) Specific questions:

15. it is not clear to me why the number of eGenes in monocytes are so much smaller than in T cells. Previous studies (Fairfax et al. 2012, Peters et al. 2016) that compared monocytes vs other cell types showed the opposite (more eGenes in monocytes than in other cell types). The authors should try to find out if any technical/analytical factors led to this discrepancy. The number of expressed genes shown in Fig. S1 are suspiciously low (they were not all on the same e.g. low performing chips, right?) and the distribution might reflect the heterogeneity of the monocyte population. Please provide literature and any additional QC plots to back up the validity of the monocyte expression data set.

The samples were randomised and those for monocytes were not on the same low performing chips. Based on the PCA of gene expression data (see figure below), the monocyte populations are more heterogeneous compared to the T-cell populations. This might contribute to the lower number of eQTLs.

Monocytes are slightly more heterogeneous, and the difference between resting and stimulated group is not as clear as that in T cells, but it's still clear that there are four distinct groups.

16. Fig. 4A: instead of just color-coding the cell type it would be more informative to show the actual PP4 for all gene-trait pairs as a color gradient in the heatmap. Supplementary figures or tables showing the PP4s of the other 3 conditions would be helpful to assess how cell type- or condition-specific these colocalizations are.

It would be difficult to show both the cell type specificity and PP4 values in Figure 4A. The PP4 values are in Supplemental Table S11 (the "PP_shared" column).

17. Please indicate if any of the vignettes have been previously found in bigger eQTL data sets.

Colocalisations observed in previous eQTL studies that are relevant with ours:

Alasoo *et al.*; macrophages from iPSCs²⁵: *CTSH* and narcolepsy and celiac disease in resting and stimulated cells; *CARD9* and Crohn's disease in IFN γ stimulated cells; *LSP1* and ulcerative colitis in resting and stimulate cells.

Kim-Hellmuth *et al.*; monocytes⁸: *AFF3* and rheumatoid arthritis in LPS-stimulated (6 hours) monocytes

Chen *et al.* BLUEPRINT (note the following coloc results are for both eQTLs and methylation QTLs)⁹: *UBE2L3* and Crohn's disease and bowel disease in monocytes, neutrophils, and T cells; *AFF3* and RA in monocytes and T cells; *BACH2* and T1D in T cells and monocytes; *CARD9* and ulcerative colitis, bowel disease, and Crohn's disease in monocytes and neutrophils; *CTSH* and T1D in monocytes; *HLA-A* and multiple sclerosis in T cells; *HLA-DRB6* and multiple sclerosis in T cells, and bowel disease and ulcerative colitis in monocytes and T cells; *UBASH3A* and rheumatoid arthritis in T cells; *UBE2L3* and bowel disease and Crohn's in monocytes, neutrophils and T cells.

4) Data availability:

Please clarify if and in which form data will be available for the research community. Ideally multiple layers of data (raw, processed, eQTL summary statistics) should be made available.

We have uploaded the full eQTL (and reQTL) summary statistics to figshare (DOI: 10.6084/m9.figshare.9585275). We are also preparing the microarray gene expression data (raw intensity data, and quantile normalised data) and we plan to make them available through ArrayExpress (<https://www.ebi.ac.uk/arrayexpress/>).

5) Minor points:

18. Fig. 1: The study design cartoon should clearly indicate that PBMCs (not monocyte and T cells) were stimulated. Additional arrows between the culture plates and the Monocyte/T cells label will help to clarify that the monocyte and T cell isolation was done after the stimulation. It might also be more correct to show only three cartoon plates instead of four as (I assume) resting adherent and non-adherent cell subsets were isolated from the same sample.

We thank the reviewer for the advice and have updated Figure 1.

19. Fig. 3:

- given the small number of trans-eQTLs, results might be easier to read in a simple table instead of a circusplot.

The circle plot emphasises the presence of *trans*-eQTLs regulating multiple *trans*-eGenes. We otherwise have the full results in multiple supplementary tables.

- are microRNAs quantifiable in these samples or were small RNAs removed (poly-dT Primer, RNA size selection) before cDNA synthesis? This information is missing in the corresponding methods section.

No, this study is not looking at microRNAs, and they are not captured by the Illumina HT12v4 microarray chips.

- Fig. S2: please provide rationale for changing the eQTL significance level to 1×10^{-5} . Are results stable at different significance thresholds?

We performed the enrichment analysis using *GARFIELD*, which accepts a single hard threshold. We have tested the enrichment at various p-value thresholds (1×10^{-3} , 1×10^{-4} , 1×10^{-5} , 1×10^{-6} , 1×10^{-7} , 1×10^{-8}), and the results are stable.

- for all suppl. Figures with colocalization regional plots (Fig. S5/6/7/10/11) I would recommend plotting only variants that are present both in the eQTL as well as in the GWAS traits, which reflect the set of variants (=intersect) that coloc uses. Currently the eQTL plots have many more variants and some GWAS traits seem to have very few data points (see Fig. S10).

Yes, currently these colocalisation regional plots show all variants in each dataset, and the numbers of available variants are different depending on the SNP chip and genotype imputation. More recent studies usually tested more genotyped and imputed variants. We deliberately chose to plot all SNPs to check if any significant SNPs (which might be additional independent causal variants), which were not available in the other dataset, were excluded from our colocalisation analysis, because this might bias the estimation of PP4.

20. Fig. S9: I might have missed it but why is the IBD/PHA T plot missing?

The number of available genetic instruments was too low to perform the Mendelian Randomisation analysis.

21. How do the HLA-C colocalization regional plots look like?

We did not observe significant colocalisation on the *HLA-C* region, the GWAS signal on this locus might not share the same causal variants with the expression of *HLA-C*. It does not affect our

conclusion from the Mendelian randomisation analysis, in which independent eQTLs of *HLA-C* were used as genetic instruments to estimate the causal effects of the gene.

22. Methods: **concentrations of LPS and PHA** used to stimulate cells are missing

We have updated the manuscript to include the concentrations of stimulants (PHA: 1 ug/ml³³; LPS: 1 ng/ml³⁴).

References:

1. Breen, M.S. *et al.* Gene expression in cord blood links genetic risk for neurodevelopmental disorders with maternal psychological distress and adverse childhood outcomes. *Brain Behav Immun* **73**, 320-330 (2018).
2. Cao-Lei, L. *et al.* DNA methylation signatures triggered by prenatal maternal stress exposure to a natural disaster: Project Ice Storm. *PLoS One* **9**, e107653 (2014).
3. Perera, F. & Herbstman, J. Prenatal environmental exposures, epigenetics, and disease. *Reprod Toxicol* **31**, 363-73 (2011).
4. Green, B.B. & Marsit, C.J. Select Prenatal Environmental Exposures and Subsequent Alterations of Gene-Specific and Repetitive Element DNA Methylation in Fetal Tissues. *Curr Environ Health Rep* **2**, 126-36 (2015).
5. Gillespie, J.H. & Turelli, M. Genotype-environment interactions and the maintenance of polygenic variation. *Genetics* **121**, 129-38 (1989).
6. Mullins, N. *et al.* Polygenic interactions with environmental adversity in the aetiology of major depressive disorder. *Psychol Med* **46**, 759-70 (2016).
7. Belsky, J. *et al.* Polygenic differential susceptibility to prenatal adversity. *Dev Psychopathol* **31**, 439-441 (2019).
8. Kim-Hellmuth, S. *et al.* Genetic regulatory effects modified by immune activation contribute to autoimmune disease associations. *Nat Commun* **8**, 266 (2017).
9. Chen, L. *et al.* Genetic Drivers of Epigenetic and Transcriptional Variation in Human Immune Cells. *Cell* **167**, 1398-1414 e24 (2016).
10. Lee, J.J. *et al.* Gene discovery and polygenic prediction from a genome-wide association study of educational attainment in 1.1 million individuals. *Nat Genet* **50**, 1112-1121 (2018).
11. Zhou, L. *et al.* Impact of human granulocyte and monocyte isolation procedures on functional studies. *Clin Vaccine Immunol* **19**, 1065-74 (2012).
12. Stegle, O., Parts, L., Piipari, M., Winn, J. & Durbin, R. Using probabilistic estimation of expression residuals (PEER) to obtain increased power and interpretability of gene expression analyses. *Nat Protoc* **7**, 500-7 (2012).
13. GTEx Consortium *et al.* Genetic effects on gene expression across human tissues. *Nature* **550**, 204-213 (2017).
14. Ye, C.J. *et al.* Intersection of population variation and autoimmunity genetics in human T cell activation. *Science* **345**, 1254665 (2014).
15. Zakharkin, S.O. *et al.* Sources of variation in Affymetrix microarray experiments. *BMC Bioinformatics* **6**, 214 (2005).
16. Bryant, P.A., Smyth, G.K., Robins-Browne, R. & Curtis, N. Technical variability is greater than biological variability in a microarray experiment but both are outweighed by changes induced by stimulation. *PLoS One* **6**, e19556 (2011).
17. Bakay, M. *et al.* Sources of variability and effect of experimental approach on expression profiling data interpretation. *BMC Bioinformatics* **3**, 4 (2002).
18. Nagalakshmi, U. *et al.* The transcriptional landscape of the yeast genome defined by RNA sequencing. *Science* **320**, 1344-9 (2008).
19. Fu, Y., Wu, P.H., Beane, T., Zamore, P.D. & Weng, Z. Elimination of PCR duplicates in RNA-seq and small RNA-seq using unique molecular identifiers. *BMC Genomics* **19**, 531 (2018).
20. Finotello, F. *et al.* Reducing bias in RNA sequencing data: a novel approach to compute counts. *BMC Bioinformatics* **15 Suppl 1**, S7 (2014).
21. Adiconis, X. *et al.* Comparative analysis of RNA sequencing methods for degraded or low-input samples. *Nat Methods* **10**, 623-9 (2013).
22. Gallego Romero, I., Pai, A.A., Tung, J. & Gilad, Y. RNA-seq: impact of RNA degradation on transcript quantification. *BMC Biol* **12**, 42 (2014).
23. Huang, Q.Q., Ritchie, S.C., Brozynska, M. & Inouye, M. Power, false discovery rate and Winner's Curse in eQTL studies. *Nucleic Acids Res* **46**, e133 (2018).
24. Urbut, S.M., Wang, G., Carbonetto, P. & Stephens, M. Flexible statistical methods for estimating and testing effects in genomic studies with multiple conditions. *Nat Genet* **51**, 187-195 (2019).

25. Alasoo, K. *et al.* Shared genetic effects on chromatin and gene expression indicate a role for enhancer priming in immune response. *Nat Genet* **50**, 424-431 (2018).
26. Davenport, E.E. *et al.* Discovering in vivo cytokine-eQTL interactions from a lupus clinical trial. *Genome Biol* **19**, 168 (2018).
27. Lissner, M.M. *et al.* Age-Related Gene Expression Differences in Monocytes from Human Neonates, Young Adults, and Older Adults. *PLoS One* **10**, e0132061 (2015).
28. Kollmann, T.R., Kampmann, B., Mazmanian, S.K., Marchant, A. & Levy, O. Protecting the Newborn and Young Infant from Infectious Diseases: Lessons from Immune Ontogeny. *Immunity* **46**, 350-363 (2017).
29. Adkins, B., Leclerc, C. & Marshall-Clarke, S. Neonatal adaptive immunity comes of age. *Nat Rev Immunol* **4**, 553-64 (2004).
30. Levy, O. Innate immunity of the newborn: basic mechanisms and clinical correlates. *Nat Rev Immunol* **7**, 379-90 (2007).
31. Makhseed, M. *et al.* Th1 and Th2 cytokine profiles in recurrent aborters with successful pregnancy and with subsequent abortions. *Hum Reprod* **16**, 2219-26 (2001).
32. Vitoratos, N. *et al.* Elevated circulating IL-1beta and TNF-alpha, and unaltered IL-6 in first-trimester pregnancies complicated by threatened abortion with an adverse outcome. *Mediators Inflamm* **2006**, 30485 (2006).
33. Heaton, T., Mallon, D., Venaille, T. & Holt, P. Staphylococcal enterotoxin induced IL-5 stimulation as a cofactor in the pathogenesis of atopic disease: the hygiene hypothesis in reverse? *Allergy* **58**, 252-6 (2003).
34. Lisciandro, J.G. *et al.* Ontogeny of Toll-like and NOD-like receptor-mediated innate immune responses in Papua New Guinean infants. *PLoS One* **7**, e36793 (2012).

Reviewers' comments:

Reviewer #1 (Remarks to the Author):

The authors have responded comprehensively to my previous comments - thank you for the helpful explanations and changes, which have enhanced the manuscript. No further comments.

Reviewer #2 (Remarks to the Author):

The authors have done substantial revisions, and the manuscript has improved. However, data supporting the authors' arguments are either not sufficient or in some cases completely missing, so my most substantial concern about data quality remains. Unfortunately, each of the five main figures still contain data that are not (yet) proven to be robust. These are described below in more detail:

Fig1:

2. Have the authors characterized the isolated cell populations (both monocytes and T cells) in terms of surface marker expression to quantify purity and cell activation status? If so, this information should be added and more importantly included as covariates in the eQTL model.

While this information was not collected at the time of experimentation, we do not expect inclusion of these as covariates to significantly change our eQTL results. The PEER factors in the eQTL model correct for latent

variation^{9,12,13} and we would expect that they would capture these differences. As far as we are aware, no previous response-eQTL study involving purified immune cells has included such information as covariates in

their eQTL model (e.g. Kim-Hellmuth et al.⁸ and Ye et al.¹⁴ as Reviewer 2 suggested in point 7).

While it is correct that PEER factors correct for hidden or technical confounders, it shows poor study design to not have cell isolation purity (or immunophenotyping) quantified for the study cohort. Thus, the authors can only describe their cell populations only as "Monocyte-enriched cultures" and "T cell-enriched cultures". This lack of quality needs to be clearly stated at the beginning of the manuscript and in the methods section to inform the reader about potential limitations of the study. The authors refer to Prescott et al. from 1999 in the method section. However, this paper isolated only T cells (monocytes have not been isolated, which is the more variable isolation protocol). T cells were isolated only from a small subset of individuals and it's not clear if any of the samples overlap between studies or if isolation protocol and quality stayed the same between 1999 and 2019. This reference should either be removed or described in more detail. If the latter it should be clearly stated that monocyte purity measurements are entirely missing.

4. Gene expression was quantified via microarray technology, which was replaced by RNA sequencing technology a long time ago and does not reflect current gold standard anymore.

We would note that there is no gold standard for gene expression analysis as choice of technology depends on sample limitations, cost, speed, analytic options, and the biological question of interest. We considered both RNA sequencing and microarrays when designing the current study. The choice of microarray technology allows for correction of technical variation and proper normalisation per-chip and per-gene. Studies have shown that the technical variation in microarray analysis can be sufficiently controlled, and the biological variation is the greatest

source of variation¹⁵⁻¹⁷. There are mature pipelines to deal with the technical issues with microarray data. RNA sequencing technology, while comprehensive, does not constitute a gold standard. It is well-known to be prone to

biases from various sources such as 3' bias¹⁸ and PCR duplication bias^{19,20}. Constructing large-scale libraries with low input RNA, as is necessary in paediatric studies, would exacerbate these artefacts and potential

introduce new ones^{21,22}.

Notably, gene expression microarrays are currently used in clinical diagnostics and clinical decisions. In comments below, Reviewer 2 also recommends that we compare our neonatal eQTLs with two adult eQTL

datasets to demonstrate replication, yet these two studies (Kim-Hellmuth et al.⁸ and Ye et al.¹⁴) also used microarray technology. While we understand the desire to utilise the latest technologies, the limitations of our paediatric cohort combined with the need to minimise technical effects meant that we had to make a conservative decision and select the most tried and robust technology available in this case.

While it is true that microarray-based eQTL studies still exist, mature pipelines to deal with technical issues of RNA-seq data are well established for a couple of years now. RNA-seq based eQTL studies exist since 2010 and have been implemented by most major eQTL studies. RNA-seq is also more widely used in the clinical setting now (Marco-Puche et.al. 2019), especially in pediatrics (Cummings et al., 2017; Cremer et al., 2017; Gonorazky et al., 2019). Given the date of other references of the study cohort (2006-2015) the authors could have acknowledged that at time of the study microarray-based expression measurements were state of the art then but not now.

Fig.2:

6. eQTL sharing analysis shown in Fig. 2B is not properly done and the resulting 74% condition- specificity is way too high. Condition-specificity cannot be simply defined as reaching an arbitrary (in this case the eigenMT-BH) p-value threshold in both conditions or not. Statistical frameworks implemented in tools such as eqtlBMA/mashr or Metasoft/Metatissue are well suited to quantify eQTL specificity and I highly recommend redoing the analysis.

We again appreciate the reviewer's recommendation, but we would note that our method of eQTL sharing analysis remains statistically rigorous. We conditioned on the most significant eQTL SNP in one condition, and applied the eQTL scan in the second condition; and we considered it as a shared eQTL if the eQTL signal in the second condition disappeared. We had to use p-value thresholds to determine if a variant was significant or not in the conditional analysis, thus thresholds from the original eQTL scan were utilised to be consistent in determining eQTLs.

That said, we also performed eQTL comparisons across four conditions using the mash model suggested by

Reviewer 2²⁴. Condition-specific eQTLs were defined as eQTLs that were active in one condition with LFSR <0.05, and were not active in the other three conditions. We observed fewer condition-specific signals using mash model: top eSNPs of 2, 38, 208, and 419 eGenes were condition-specific eQTLs in resting monocytes, stimulated monocytes, resting T cells, and stimulated T cells, respectively, while our conditional analyses identified 24, 151, 504, and 887 condition-specific eQTLs. All of the condition-specific signals identified in monocytes using mashr were replicated in our conditional analysis, and 204 and 413 signals in resting and stimulated T cells, respectively, were identified using both methods. However, it is not possible to know definitively whether this is because mash has lesser power or conditional analysis has a higher false discovery rate. To investigate this further, we looked at several factors. Firstly, the hierarchical multiple testing correction method used in our study to determine significant eQTLs was more stringent than the mash model (mashr does not account for multiple testing), and there were many cases where genes had active eQTL signals determined by mashr, but the same genes did not have any significant eQTLs using our multiple testing correction. Secondly, the mash model focused on top SNPs per gene, thus it missed secondary eQTLs for genes with multiple independent eQTLs. Lastly, the mash model does not distinguish between causal eQTLs from those that are in LD with the causal variants, while this was better accounted for in the conditional analysis.

Please add above mentioned mashr results as supplementary figure (e.g. as Venn diagrams) to provide readers the range of condition-specific eQTLs depending on which method you use. Higher concordance rates would have been desirable but at least this way the reader can set the numbers of former Fig. 2B (which is now supplementary fig. S2) in context.

As a comment: mashr jointly analyses the sign and magnitude of an eQTL across multiple conditions to estimate if an eQTL 1) is active and 2) if it's shared or specific to a single condition. That is a major advantage as it allows to identify eQTLs in studies with limited power but multiple measurements (which the authors falsely claim as being less stringent) and gives a more precise estimate of the sharing pattern as it considers the magnitude across conditions as well. This is nicely demonstrated in this study: even though mashr detects many more eQTLs than their method (according to the authors) the number of condition-specific eQTLs is up to 10x smaller in mashr indicating that most of the eQTLs presented as being condition-specific in this study actually show evidence of activity in other conditions. Secondly, Mashr analysis doesn't need to be limited to the top SNP per gene. The original paper used the top SNP per gene, but in theory secondary eQTLs (that are not in LD with the primary eQTL) could certainly have been included. Lastly it is unclear to me how the authors' conditional analysis (without finemapping) accounts for causal eQTLs better if the top variant in their analysis is the same that is provided to mashr.

8. reQTL mapping: while the interaction model is reasonable the corresponding permutation scheme is not valid as the bond between genotype and expression still remains (see Buzkova et al. *Annals of Human Genetics* 2010, and Buzkova, *Epidemiol. Methods* 2016 for explanation and solutions). Did the authors check if p-values are calibrated? Permutation schemes for interaction terms are non-trivial (e.g. parametric bootstrap) and one could consider applying a very conservative correction instead (Bonferroni or more stringent). With the current permutation scheme the proportion of reQTLs in Fig.2A are highly overestimated and basically useless in the current form. Also, Table S7+8 do not provide the lmer nominal pvalues (only permutation p is listed).

We would acknowledge that the rationale for the permutation testing of response eQTLs could have been clearer. In contrast to regular eQTL mapping (no interactions), it is in fact desirable to maintain the bond between

genotype and expression during permutation when detecting response eQTLs^{5,26}. In the reQTL mapping, we tested if the changes in eQTL effects across conditions were significant, given that genotype had effects on gene expression. Also, the papers and the parametric bootstrap method mentioned by reviewer 2 are actually trying to retain the genotype-expression correlation (or, the "main effects" in the papers).

We used the same method with the Alasoo et al. study²⁵. Permutations were performed to estimate empirical p-values (not to correct for multiple testing), and we applied BH FDR on those empirical p-values. We do not believe that the proportion of reQTLs was overestimated.

Please provide plots of calibrated p-values of the permutation scheme. Just because the permutation method has been used in another study does not automatically prove that the method is robust. Also, Table S7+8 still don't include Imer nominal p-values.

Fig.3:

9. trans-eQTL analysis is still challenging and is not recommended for studies with low sample size. Without evidence of replication it is difficult to assess how valid the findings of this study are.

Indeed, our study is not well-powered for trans-eQTL detection. However, the trans-eQTLs are still informative as long as multiple testing is properly controlled. We applied the genome-wide FDR method used in the GTEx project, which also mapped trans-eQTLs in tissues with sample sizes ranging from 79 to 361¹³.

Reviewer 2 commented that one of our trans-eQTL on the RPS26 locus was not a novel finding because previous studies have already reported this locus. We counter that it is evidence of replication and method validity, and remains scientifically-valuable.

Having one previously described trans eQTL (RPS26) among the list of presented trans eQTLs can hardly be called replication. Many highly powered (adult) eQTL data sets exist for blood. Without evidence of replication trans eQTL data should either be moved to the supplement or removed entirely.

19. Fig. 3:

- are microRNAs quantifiable in these samples or were small RNAs removed (poly-dT Primer, RNA size selection) before cDNA synthesis? This information is missing in the corresponding methods section.

No, this study is not looking at microRNAs, and they are not captured by the Illumina HT12v4 microarray chips.

If microRNAs are not captured what are your trans eGenes MIR130A, MIR330 and MIR 1471 resemble?

-Fig. S2: please provide rationale for changing the eQTL significance level to 1×10^{-5} . Are results stable at different significance thresholds?

We performed the enrichment analysis using GARFIELD, which accepts a single hard threshold. We have tested the enrichment at various p-value thresholds (1×10^{-3} , 1×10^{-4} , 1×10^{-5} , 1×10^{-6} , 1×10^{-7} , 1×10^{-8}), and the results are stable.

Please provide results of using different p-value thresholds as supplementary tables or figure that shows the consistency.

Fig. 4:

0. colocalization analysis: coloc is very sensitive to the prior selection. Modifying just one prior often doesn't solve the problem. Did the authors check if posterior probabilities of their loci stay high across a range of different p1, p2 and p12 priors? I would recommend running this analysis (e.g. applying bayesian model averaging) or optionally using more advanced colocalization methods such as eCAVIAR or enloc.

We agree that coloc is sensitive to the prior selection; that is why we used a more stringent prior to be conservative. We have also performed sensitivity analysis using the sensitivity() function that has been implemented in the new version of coloc, which evaluates if the colocalisation is robust across a range of

different p12 prior values ranging from 1×10^{-9} to 1×10^{-5} . All our significant colocalisations passed the sensitivity

analysis. We also varied p_1 and p_2 priors from 1×10^{-4} to 1×10^{-5} , and the posterior probabilities of colocalisations stayed high.

Please add plots/results of the sensitivity analysis and p_1 , p_2 prior variation as supplementary figures/tables.

16. Fig. 4A: instead of just color-coding the cell type it would be more informative to show the actual PP4 for all gene-trait pairs as a color gradient in the heatmap. Supplementary figures or tables showing the PP4s of the other 3 conditions would be helpful to assess how cell type- or condition- specific these colocalizations are.

It would be difficult to show both the cell type specificity and PP4 values in Figure 4A. The PP4 values are in Supplemental Table S11 (the "PP_shared" column).

I think you are referring to Table S10. "PP_shared" shows the PP4 of only the colocalizing cell type condition but not the other ones. Please provide "PP_shared" for all 4 conditions in separate columns next to each other to enable the direct comparison of PP4 across conditions. Without this information it is not possible to assess if e.g. a T cell-specific colocalization is really specific (PP4 in other conditions being close to 0) or if it's just a threshold issue with PP4 in other conditions being close to the T cell PP4.

17. Please indicate if any of the vignettes have been previously found in bigger eQTL data

sets. Colocalisations observed in previous eQTL studies that are relevant with ours:

Alasoo et al.; macrophages from iPSCs²⁵: CTSH and narcolepsy and celiac disease in resting and stimulated cells; CARD9 and Crohn's disease in IFN γ stimulated cells; LSP1 and ulcerative colitis in resting and stimulate cells.

Kim-Hellmuth et al.; monocytes⁸: AFF3 and rheumatoid arthritis in LPS-stimulated (6 hours) monocytes

Chen et al. BLUEPRINT (note the following coloc results are for both eQTLs and methylation QTLs)⁹: UBE2L3 and Crohn's disease and bowel disease in monocytes, neutrophils, and T cells; AFF3 and RA in monocytes and T cells; BACH2 and T1D in T cells and monocytes; CARD9 and ulcerative colitis, bowel disease, and Crohn's disease in monocytes and neutrophils; CTSH and T1D in monocytes; HLA-A and multiple sclerosis in T cells; HLA-DRB6 and multiple sclerosis in T cells, and bowel disease and ulcerative colitis in monocytes and T cells; UBASH3A and rheumatoid arthritis in T cells; UBE2L3 and bowel disease and Crohn's in monocytes, neutrophils and T cells.

Please ensure that this information is incorporated in the manuscript.

- for all suppl. Figures with colocalization regional plots (Fig. S5/6/7/10/11) I would recommend plotting only variants that are present both in the eQTL as well as in the GWAS traits, which reflect the set of variants (=intersect) that coloc uses. Currently the eQTL plots have many more variants and some GWAS traits seem to have very few data points (see Fig. S10).

Yes, currently these colocalisation regional plots show all variants in each dataset, and the numbers of available variants are different depending on the SNP chip and genotype imputation. More recent studies usually tested more genotyped and imputed variants. We deliberately chose to plot all SNPs to check if any significant SNPs (which might be additional independent causal variants), which were not available in the other dataset, were excluded from our colocalisation analysis, because this might bias the estimation of PP4.

This is not the correct illustration of coloc results. Since coloc posterior probabilities are only based on the intersect of the two data sets, please provide the regional plots accordingly. These can be provided as additional supplementary figures to the current ones if the authors feel that the current figures are necessary.

Fig. 5:

20. Fig. S9: I might have missed it but why is the IBD/PHA T plot missing?

The number of available genetic instruments was too low to perform the Mendelian Randomisation analysis.

The figure legend should state why the IBD/PHA T plot is missing

21. How do the HLA-C colocalization regional plots look like?

We did not observe significant colocalisation on the HLA-C region, the GWAS signal on this locus might not share the same causal variants with the expression of HLA-C. It does not affect our conclusion from the Mendelian randomisation analysis, in which independent eQTLs of HLA-C were used as genetic instruments to estimate the causal effects of the gene.

Even if there is no significant colocalization, corresponding regional plots of a main figure (Fig. 5) should be provided to ensure transparency of underlying data. All information should be available for both vignettes (BTN3A2 and HLA-C) to the reader to assess the validity and strength of the putative causal effect of HLA-C in autoimmunity. If independent eQTLs of HLA-C have been used as genetic instruments, regional plot of conditional eQTLs (conditioned for the top eQTL) could be provided. If regional association plots still look extremely different the authors should add a discussion section to assist the interpretation of such a result.

Reviewer #3 (Remarks to the Author):

In my opinion, Huang et al. and collaborators have addressed properly the concerns about the previous version of the manuscript pointed out by reviewer#2.

The details about the selected monocyte isolation protocol have been supported with adequate justifications. However, although cell purity is not normally included as covariate in QTL studies, it is presented as quality control for the reported datasets. Would it be possible for authors to calculate it from stored or pooled material?

The use of expression microarrays is also well justified. It should be noted that RNA-seq would allow the authors to perform additional analyses such as identification of isoform QTLs, which could be especially relevant in neonatal tissues. However, despite the limitations as the authors point out microarrays are robust platforms to study gene expression and normalization methods are well-established. Authors should share their QC and normalization pipeline details.

I consider that the QTL analysis methods have been extended and fulfil current gold-standards. The multiple-testing p-value correction methods and thresholds are also in the standards of the field.

The analysis of specific trans-eQTLs, not trans eQTLs at the genome-wide level is properly justified.

Although the overlap with previous studies in adults was addressed properly using the mash model. Pi1 values, as requested by the previous reviewer, would also be informative and would allow the reader (and the reviewers) to compare the overlap with previous studies in the context of previous comparisons between cell types and stimulation strategies.

Minor points:

- 1) It should be clearly specified how many individuals passed QC for all the different conditions and cell types.
- 2) What was the R² and D' between colocalizing lead SNP variants for the eQTLs and the immune GWAS SNPs? This is important to confirm that both eQTLs and GWAS are really reflecting the same signal.

Reviewer #3:

In my opinion, Huang et al. and collaborators have addressed properly the concerns about the previous version of the manuscript pointed out by reviewer#2.

We thank the reviewer for their time in providing additional assessment.

1. The details about the selected monocyte isolation protocol have been supported with adequate justifications. However, although cell purity is not normally included as covariate in QTL studies, it is presented as quality control for the reported datasets. Would it be possible for authors to calculate it from stored or pooled material?

While we do not have remaining neonatal material to test cell purity, we have endeavoured to perform *in silico* analyses to estimate the abundances of relevant cell types from the CAS gene expression data using CIBERSORTx (<https://cibersortx.stanford.edu>). We have added the results as **Figure S1**. While there are limitations to using computational methods to infer cell purity, we found that the vast majority of transcriptomic signatures from neonatal samples were dominated by the intended cell types. The LPS-stimulated monocyte/macrophage-enriched cultures were estimated to have a median of 82.9% myeloid cells. The PHA-stimulated T-cell cultures were estimated to have a median of 82.7% T cells. For our purposes here, we would assume that these estimates represent lower bounds as CIBERSORTx, and other computational cell abundance tools, will likely have lower detection power for neonatal forms of these cell types.

Furthermore, we assessed whether our existing analyses using PEER factors already capture the inferred cell type abundances. The PEER factors in macrophage/monocyte-enriched cultures showed strong associations with the inferred myeloid cell abundances in both resting cultures, e.g. PEER1 (Pearson correlation: 0.47) and PEER2 (Pearson: -0.78), and in LPS-stimulated cultures, e.g. PEER3 (Pearson: -0.36), PEER2 (Pearson: -0.22), and PEER9 (Pearson: -0.20). Similarly, while cell purity was greater and more homogeneous as expected, the PEER factors for T-cell cultures also captured differences in inferred T cell abundance: PEER9 (Pearson: 0.22) in resting and PEER1 (Pearson: 0.18) in PHA-stimulated samples.

Taken together, we are confident that our isolation protocol has yielded high fidelity samples and that our statistical models have been appropriately controlled for differences in cell abundances between samples.

2. The use of expression microarrays is also well justified. It should be noted that RNA-seq would allow the authors to perform additional analyses such as identification of isoform QTLs, which could be especially relevant in neonatal tissues. However, despite the limitations as the authors point out microarrays are robust platforms to study gene expression and normalization methods are well-established. Authors should share their QC and normalization pipeline details.

We thank the reviewer for the suggestion. We have now added a detailed pipeline figure of the QC and microarray data processing to the Supplementary Materials (**Figure S16**).

3. Although the overlap with previous studies in adults was addressed properly using the mash model. π_1 values, as requested by the previous reviewer, would also be informative and would allow the reader (and the reviewers) to compare the overlap with previous studies in the context of previous comparisons between cell types and stimulation strategies.

We agree with the reviewer and now also report the replication rate π_1 value. In terms of comparison, the π_1 statistic and the mash model provide different ways to address the

question with both being informative. π_1 values were high (0.88 for resting T cells, and >0.9 for myeloid cells), with somewhat greater overlap observed between neonatal myeloid cells and adult monocytes, consistent with the MashR results (**Figure 2**).

Minor points:

1) It should be clearly specified how many individuals passed QC for all the different conditions and cell types.

We thank the reviewer for the suggestion and have now added this information in the first paragraph of the Results section (pg 4). Amongst the 135 genotyped neonates, 106 had transcriptomes passing QC for both resting and stimulated conditions of monocyte/macrophage-enriched cultures, and a corresponding number of 119 for T-cell-enriched cultures. Ninety-five had post-QC data for all four experimental conditions.

2) What was the R^2 and D' between colocalizing lead SNP variants for the eQTLs and the immune GWAS SNPs? This is important to confirm that both eQTLs and GWAS are really reflecting the same signal.

We thank the reviewer for the suggestion and have now added this information to **Table S10**. As expected, most lead SNPs for the two signals are highly correlated with a few cases where the R^2 is low but D' is high, a well-known case when there is a difference in allele frequencies between the two lead SNPs.

Reviewer #2

2. While it is correct that PEER factors correct for hidden or technical confounders, it shows poor study design to not have cell isolation purity (or immunophenotyping) quantified for the study cohort. Thus, the authors can only describe their cell populations only as “Monocyte-enriched cultures” and “T cell-enriched cultures”. This lack of quality needs to be clearly stated at the beginning of the manuscript and in the methods section to inform the reader about potential limitations of the study. The authors refer to Prescott et al. from 1999 in the method section. However, this paper isolated only T cells (monocytes have not been isolated, which is the more variable isolation protocol). T cells were isolated only from a small subset of individuals and it's not clear if any of the samples overlap between studies or if isolation protocol and quality stayed the same between 1999 and 2019. This reference should either be removed or described in more detail. If the latter it should be clearly stated that monocyte purity measurements are entirely missing.

As suggested, we have now clearly stated that cell purity was not experimentally confirmed in Results and Methods (pg4, paragraph 1; pg11, Methods paragraph 1) as well as incorporated the language the reviewer suggests to describe the cell cultures. We have rewritten the first paragraph in the Methods (pg 11) regarding the Prescott *et al.* reference to clarify that this paper refers specially to T-cell preparation, and to indicate that the methodology for T-cell enrichment in our study was precisely the same. In the modified paragraph, we have also described the method for myeloid cell enrichment, and the use of *in silico* analyses to estimate the abundances of relevant cell types (see Reviewer #3 point 1).

4. While it is true that microarray-based eQTL studies still exists, mature pipelines to deal with technical issues of RNA-seq data are well established for a couple of years now. RNA-seq based eQTL studies exist since 2010 and have been implemented by most major eQTL

studies. RNA-seq is also more widely used in the clinical setting now (Marco-Puche et al. 2019), especially in pediatrics (Cummings et al., 2017; Cremer et al., 2017; Gonorazky et al., 2019). Given the date of other references of the study cohort (2006-2015) the authors could have acknowledged that at time of the study microarray-based expression measurements were state of the art then but not now.

While we disagree with the reviewer on gold-standard consensus and the suitability of RNA-seq for newborn sample volumes, we have added a discussion of the advantages and disadvantages of microarrays and RNA-seq technology to the manuscript (last paragraph on pg 10).

6. Please add above mentioned mashr results as supplementary figure (e.g. as Venn diagrams) to provide readers the range of condition-specific eQTLs depending on which method you use. Higher concordance rates would have been desirable but at least this way the reader can set the numbers of former Fig. 2B (which is now supplementary fig. S2) in context.

As a comment: mashr jointly analyses the sign and magnitude of an eQTL across multiple conditions to estimate if an eQTL 1) is active and 2) if it's shared or specific to a single condition. That is a major advantage as it allows to identify eQTLs in studies with limited power but multiple measurements (which the authors falsely claim as being less stringent) and gives a more precise estimate of the sharing pattern as it considers the magnitude across conditions as well. This is nicely demonstrated in this study: even though mashr detects many more eQTLs than their method (according to the authors) the number of condition-specific eQTLs is up to 10x smaller in mashr indicating that most of the eQTLs presented as being condition-specific in this study actually show evidence of activity in other conditions. Secondly, Mashr analysis doesn't need to be limited to the top SNP per gene. The original paper used the top SNP per gene, but in theory secondary eQTLs (that are not in LD with the primary eQTL) could certainly have been included. Lastly it is unclear to me how the authors' conditional analysis (without finemapping) accounts for causal eQTLs better if the top variant in their analysis is the same that is provided to mashr.

We thank the reviewer for the suggestion. We have added Venn diagrams showing the above mashr results to **Figure S3B**.

8. Please provide plots of calibrated p-values of the permutation scheme. Just because the permutation method has been used in another study does not automatically prove that the method is robust. Also, Table S7+8 still don't include lmer nominal p-values.

To clarify, there are no "calibrated" p-values because we are estimating the p-values directly with permutations. The lme4 R package does not provide nominal p-values like a normal linear regression model, thus we used permutations to calculate the p-values. Previously, lme4 used MCMC to calculate p-values, but it was not reliable when the variances of the estimated random effects were small, so it was withdrawn (see: <https://rdrr.io/cran/lme4/man/pvalues.html>). Our analysis choice was therefore to utilise permutations since it is well established as both robust and widely applied.

9. Having one previously described trans eQTL (RPS26) among the list of presented trans eQTLs can hardly be called replication. Many highly powered (adult) eQTL data sets exist for blood. Without evidence of replication trans eQTL data should either be moved to the supplement or removed entirely.

We appreciate the reviewer's comment regarding the *trans* eQTL analysis. We have moderated our language and appropriately caveat this analysis in both the Results (pg 5,

paragraph 4) and the Discussion (in particular pg 11, paragraph 1) and have emphasised the need for larger, more powerful studies.

19. If microRNAs are not captured what are your trans eGenes MIR130A, MIR330 and MIR 1471 resemble?

We can clarify that small RNA probes passing QC were included in the analysis.

-Fig. S2: Please provide results of using different p-value thresholds as supplementary tables or figure that shows the consistency.

We thank the reviewer for the advice and have added the results of using different eQTL significance levels to **Figure S4**. The enrichment patterns are indeed very similar across these thresholds.

10. colocalization analysis: Please add plots/results of the sensitivity analysis and p1, p2 prior variation as supplementary figures/tables.

We have added additional columns to **Table S10** to include (1) results of sensitivity analysis, and (2) results of using different p1 and p2 priors.

16. I think you are referring to Table S10. "PP_shared" shows the PP4 of only the colocalizing cell type condition but not the other ones. Please provide "PP_shared" for all 4 conditions in separate columns next to each other to enable the direct comparison of PP4 across conditions. Without this information it is not possible to assess if e.g. a T cell-specific colocalization is really specific (PP4 in other conditions being close to 0) or if it's just a threshold issue with PP4 in other conditions being close to the T cell PP4.

We thank the reviewer for the suggestion, and we agree that this information is very useful. We have added four columns to **Table S10** as the reviewer suggested.

17. This is not the correct illustration of coloc results. Since coloc posterior probabilities are only based on the intersect of the two data sets, please provide the regional plots accordingly. These can be provided as additional supplementary figures to the current ones if the authors feel that the current figures are necessary.

We thank the reviewer's suggestion, and we have updated the eQTL plots in **Figure S12** to show SNPs that were tested in the colocalisation analysis. All GWAS regional plots show the intersect of the two data sets. The majority of recent GWASs used imputed genotype data (except the GWAS in **Figure S12**) so the difference between the two sets is minimal.

20. The figure legend should state why the IBD/PHA T plot is missing

We thank the reviewer and have updated the legend for **Figure S11**.

21. Even if there is no significant colocalization, corresponding regional plots of a main figure (Fig. 5) should be provided to ensure transparency of underlying data. All information should be available for both vignettes (BTN3A2 and HLA-C) to the reader to assess the validity and strength of the putative causal effect of HLA-C in autoimmunity. If independent eQTLs of HLA-C have been used as genetic instruments, regional plot of conditional eQTLs (conditioned for the top eQTL) could be provided. If regional association plots still look extremely different the authors should add a discussion section to assist the interpretation of such a result.

We thank the reviewer for the suggestion, and have added the regional plots (**Figure S13–14**) for *HLA-C*. The GWAS signals on the *HLA-C* locus are composed of many SNPs that show extremely significance levels.

REVIEWERS' COMMENTS:

Reviewer #2 (Remarks to the Author):

The points have been addressed except for two minor points that remain. I trust the authors to address them appropriately.

1. the main text on page 4 still says "The majority (74%) of eQTL signals were specific to one cell type or stimulatory condition (Figure S3),..." without appreciating the new results shown in Fig. S3b that strongly indicate that 74% is most probably overestimated.

The fact that only 10-50% of the ones they find to be cell type- or condition-specific are replicated with an independent method should be specifically mentioned in the main text.

2. I had asked the authors to incorporate the information on which of their GWAS vignettes have already been published to avoid the appearance of novelty. Unfortunately, I cannot find corresponding references on page 7 of the main text. So, for those vignettes that are already published (e.g. BACH2, CTSH, etc.) please indicate that with a sentence and the corresponding reference.

I am copying my previous request here for reference:

17. Please indicate if any of the vignettes have been previously found in bigger eQTL data sets. Colocalisations observed in previous eQTL studies that are relevant with ours:

Alasoo et al.; macrophages from iPSCs²⁵: CTSH and narcolepsy and celiac disease in resting and stimulated cells; CARD9 and Crohn's disease in IFN γ stimulated cells; LSP1 and ulcerative colitis in resting and stimulate cells.

Kim-Hellmuth et al.; monocytes⁸: AFF3 and rheumatoid arthritis in LPS-stimulated (6 hours) monocytes

Chen et al. BLUEPRINT (note the following coloc results are for both eQTLs and methylation QTLs)⁹: UBE2L3 and Crohn's disease and bowel disease in monocytes, neutrophils, and T cells; AFF3 and RA in monocytes and T cells; BACH2 and T1D in T cells and monocytes; CARD9 and ulcerative colitis, bowel disease, and Crohn's disease in monocytes and neutrophils; CTSH and T1D in monocytes; HLA-A and multiple sclerosis in T cells; HLA-DRB6 and multiple sclerosis in T cells, and bowel disease and ulcerative colitis in monocytes and T cells; UBASH3A and rheumatoid arthritis in T cells; UBE2L3 and bowel disease and Crohn's in monocytes, neutrophils and T cells.

Please ensure that this information is incorporated in the manuscript. not addressed in the second revision

Reviewer #3 (Remarks to the Author):

I consider that the authors have properly addressed my comments to the previous version of the manuscript.

REVIEWERS' COMMENTS:

Reviewer #2 (Remarks to the Author):

The points have been addressed except for two minor points that remain. I trust the authors to address them appropriately.

1. the main text on page 4 still says “The majority (74%) of eQTL signals were specific to one cell type or stimulatory condition (Figure S3),...” without appreciating the new results shown in Fig. S3b that strongly indicate that 74% is most probably overestimated. The fact that only 10-50% of the ones they find to be cell type- or condition-specific are replicated with an independent method should be specifically mentioned in the main text.

We thank the reviewer. We have now specifically mentioned that about 10–50% of the results were replicated with the new method and cited Supplementary Figure 3B in the last paragraph on page 4.

2. I had asked the authors to incorporate the information on which of their GWAS vignettes have already been published to avoid the appearance of novelty. Unfortunately, I cannot find corresponding references on page 7 of the main text. So, for those vignettes that are already published (e.g. BACH2, CTSH, etc.) please indicate that with a sentence and the corresponding reference.

I am copying my previous request here for reference:

17. Please indicate if any of the vignettes have been previously found in bigger eQTL data sets.

Colocalisations observed in previous eQTL studies that are relevant with ours:

Alasoo et al.; macrophages from iPSCs²⁵: CTSH and narcolepsy and celiac disease in resting and stimulated cells; CARD9 and Crohn’s disease in IFN γ stimulated cells; LSP1 and ulcerative colitis in resting and stimulate cells.

Kim-Hellmuth et al.; monocytes⁸: AFF3 and rheumatoid arthritis in LPS-stimulated (6 hours) monocytes

Chen et al. BLUEPRINT (note the following coloc results are for both eQTLs and methylation QTLs)⁹: UBE2L3 and Crohn’s disease and bowel disease in monocytes, neutrophils, and T cells; AFF3 and RA in monocytes and T cells; BACH2 and T1D in T cells and monocytes; CARD9 and ulcerative colitis, bowel disease, and Crohn’s disease in monocytes and neutrophils; CTSH and T1D in monocytes; HLA-A and multiple sclerosis in T cells; HLA-DRB6 and multiple sclerosis in T cells, and bowel disease and ulcerative colitis in monocytes and T cells; UBASH3A and rheumatoid arthritis in T cells; UBE2L3 and bowel disease and Crohn’s in monocytes, neutrophils and T cells.

Please ensure that this information is incorporated in the manuscript. not addressed in the second revision

We thank the reviewer for the suggestion. We have now incorporated the relevant information accordingly on page 7:

“Our analysis replicated the colocalisation of the cis-eQTL for BACH2 in resting T cells with variants for type 1 diabetes²⁷, and also revealed widespread novel colocalisation with autoimmune thyroid disease, celiac disease, multiple sclerosis, rheumatoid arthritis.”

“Cis-eQTLs for CTSH in resting myeloid cells and resting T cells both colocalised with GWAS hits for celiac disease, narcolepsy, and type 1 diabetes, which has been observed in immune cell types from adults: colocalisation with causal variants of celiac disease and narcolepsy was reported in macrophages¹⁴, and type 1 diabetes in adult monocytes²⁷.”

The colocalisation of eQTLs for *IL13* and *CCL20* (page 7) were novel.

Reviewer #3 (Remarks to the Author):

I consider that the authors have properly addressed my comments to the previous version of the manuscript.